# Ca$^{2+}$-mediated higher-order assembly of heterodimers in amino acid transport system b$^{0,+}$ biogenesis and cystinuria

Yongchan Lee [1,2,5✉], Pattama Wiriyasermkul[3,4,5], Pornparn Kongpracha[3,4], Satomi Moriyama[4], Deryck J. Mills[1,6], Werner Kühlbrandt [1] & Shushi Nagamori [3,4✉]

Cystinuria is a genetic disorder characterized by overexcretion of dibasic amino acids and cystine, causing recurrent kidney stones and kidney failure. Mutations of the regulatory glycoprotein rBAT and the amino acid transporter b$^{0,+}$AT, which constitute system b$^{0,+}$, are linked to type I and non-type I cystinuria respectively and they exhibit distinct phenotypes due to protein trafficking defects or catalytic inactivation. Here, using electron cryo-microscopy and biochemistry, we discover that Ca$^{2+}$ mediates higher-order assembly of system b$^{0,+}$. Ca$^{2+}$ stabilizes the interface between two rBAT molecules, leading to super-dimerization of b$^{0,+}$AT–rBAT, which in turn facilitates N-glycan maturation and protein trafficking. A cystinuria mutant T216M and mutations of the Ca$^{2+}$ site of rBAT cause the loss of higher-order assemblies, resulting in protein trapping at the ER and the loss of function. These results provide the molecular basis of system b$^{0,+}$ biogenesis and type I cystinuria and serve as a guide to develop new therapeutic strategies against it. More broadly, our findings reveal an unprecedented link between transporter oligomeric assembly and protein-trafficking diseases.

[1] Department of Structural Biology, Max Planck Institute of Biophysics, 60438 Frankfurt, Germany. [2] Graduate School of Medical Life Science, Yokohama City University, Kanagawa 230-0045, Japan. [3] Department of Laboratory Medicine, The Jikei University School of Medicine, Tokyo 105-8461, Japan. [4] Department of Collaborative Research for Bio-Molecular Dynamics, Nara Medical University, Nara 634-8521, Japan. [5] These authors contributed equally: Yongchan Lee, Pattama Wiriyasermkul. [6] Deceased: Deryck J. Mills. ✉email: yongchan.lee@biophys.mpg.de; snagamori@nagamori-lab.jp

Cystinuria is an inherited disorder in which defective amino acid reabsorption in the kidney causes over-excretion of cystine and dibasic amino acids into urine[1]. Due to the poor solubility of cystine, patients suffer from recurrent kidney stones, which can grow to several centimeters, causing acute pain and kidney failure[2]. Many patients thus require life-long care, and studies estimate that 1 in 7000 newborns are affected by the disease[1]. Common treatments include dietary therapy[3], extracorporeal shock wave lithotripsy, management of cystine dilution[4] and open surgery, but so far no breakthrough has been made beyond these symptomatic measures due to an incomplete understanding of the disease mechanisms.

Cystinuria is caused by mutations in system $b^{0,+}$, which is a $Na^+$-independent cystine/dibasic amino acid exchanger localized at brush border cells in the kidney and intestine. System $b^{0,+}$ is composed of two subunits, a single-transmembrane glycoprotein rBAT[5] (also known as D2, NBAT and SLC3A1) and a 12-transmembrane transporter $b^{0,+}AT$[6] (also known as BAT1 and SLC7A9), which belong to the heteromeric amino acid transporter (HAT) family[7]. A recent study revealed that rBAT associates with another SLC7 member, AGT1 (SLC7A13), which shows distinct proximal tubular localization compared to $b^{0,+}AT$[8]. Although recent structural elucidation of the prototypical HAT transporter, LAT1–CD98hc, has shown how two subunits form a tight heterodimeric assembly[9,10], the low sequence identity of rBAT to CD98hc (28% in the aligned region) and the insertion of a total of ~100 residues in domain B loops and the C-terminal region[11] have precluded a detailed molecular understanding of $b^{0,+}AT$–rBAT.

Clinical and biochemical studies have shown that mutations of the two subunits are linked to different phenotypes, known as type I cystinuria for rBAT and non-type I for $b^{0,+}AT$[1]. Whereas most type I mutations cause system $b^{0,+}$ malfunction[12], non-type I mutations abolish the transport activity itself[13]. In addition, while all characterized type I mutations show the disease phenotype only in homozygotes, some non-type I mutations can trigger the disease even in heterozygous individuals[14], a phenomenon that currently lacks a clear explanation. Recently, the cryo-EM structures of human $b^{0,+}AT$–rBAT in detergent revealed its architecture and suggested a mechanism of amino acid antiport and its dysfunctions[15,16]. However, these structures did not explain why certain type I mutations cause protein trafficking defects.

To better understand the molecular mechanisms of system $b^{0,+}$ biogenesis and cystinuria, we study the structure and function of ovine system $b^{0,+}$. We determine the electron cryo-microscopy (cryo-EM) structure of ovine $b^{0,+}AT$–rBAT in lipid nanodiscs, which unveils the super-dimeric $(b^{0,+}AT-rBAT)_2$ complex embedded in a curved lipid bilayer. We identify a $Ca^{2+}$-binding site in the rBAT ectodomain, and demonstrate that this site is essential for higher-order assembly of system $b^{0,+}$ in the ER, which in turn facilitates its N-glycan maturation and trafficking to the plasma membrane. Furthermore, we discover that the loss of higher-order assembly is correlated with some type I mutations, revealing a previously unknown link between protein oligomerization and the disease. These results shed light on the molecular mechanisms of type I cystinuria.

## Results

**Mammalian $b^{0,+}AT$–rBAT complexes form a super-dimer.** We purified ovine and murine $b^{0,+}AT$–rBAT complexes, which share 80–88% sequence identities with the human counterparts (Supplementary Fig. 1) and showed good behavior in detergent solution (Supplementary Fig. 2a). SDS-PAGE analyses indicated a high-molecular weight band at the top (marked as super-dimer) and a heterodimer band (~130 kDa) under non-reducing

conditions (Supplementary Fig. 2b), which separated into two bands for $b^{0,+}AT$ and rBAT under reducing conditions, confirming the formation of a disulfide-linked complex. Negative-stain electron microscopy showed two rBAT ectodomains arranged head-to-head, indicative of a higher-order assembly (Fig. 1a). In addition, each ectodomain of rBAT is substantially larger than that of LAT1–CD98hc (Fig. 1b), reflecting the ~100 amino acid insertion. To verify the functional integrity of the purified complex, we reconstituted ovine $b^{0,+}AT$–rBAT into liposomes (Supplementary Fig. 2d) and measured amino acid transport activities (Fig. 1c). $b^{0,+}AT$–rBAT showed significant L-[$^3$H]Arg uptake when liposomes were loaded with L-Arg. Transport activity was negligible when the liposomes were not loaded with L-Arg, confirming the L-Arg/L-Arg exchange activity (Fig. 1c).

Initial single-particle cryo-EM analysis of ovine $b^{0,+}AT$–rBAT complex in detergent yielded a map at an overall resolution of 3.9 Å (Supplementary Fig. 3a). After rigorous optimization, we reconstituted the complex into lipid nanodiscs consisting of MSP1E3D1, phospholipids and cholesterol (Supplementary Fig. 2c, see Methods for detail), which yielded a better map at 2.9 Å global resolution (Supplementary Fig. 3b–f). To further improve the map quality, we performed the multi-body refinement and focused refinement[17], improving the map resolution to 2.6 Å for the extracellular domain and 3.0 Å for the individual heterodimer (Supplementary Fig. 3e,f). These maps allowed us to build an atomic model of ovine $b^{0,+}AT$–rBAT (Supplementary Figs. 4a–e, 5).

**Overall structure of the ovine $b^{0,+}AT$–rBAT complex.** The overall structure of the ovine system $b^{0,+}$ shows two copies of $b^{0,+}AT$ and rBAT, arranged as a super-dimer of the heterodimer (Fig. 1d–e). This architecture is essentially identical to the recent structures of human $b^{0,+}AT$–rBAT[15], confirming a conserved higher-order assembly (Supplementary Fig. 6b). rBAT has three domains, namely the ectodomain, a single transmembrane helix (TM1') and the cytoplasmic N-terminal helix (NH) (Fig. 2a–c), in addition to 62 unresolved disordered residues at the N-terminus. The ectodomain displays a glucosidase-like fold[11,18], which consists of domains A, B, and C (Fig. 2b). The cryo-EM map also resolved a $Ca^{2+}$ ion and six N-linked glycans for each rBAT molecule, as discussed later in detail.

$b^{0,+}AT$ shares structural and sequence homology with the APC superfamily and shows a typical LeuT-fold (Fig. 2c), consisting of 12 transmembrane helices (TM1–12) that contain 5 + 5 TM inverted repeats[19]. Other structural features involve two helices in the extracellular loop 4 (EL4a and EL4b), one helix in intracellular loop 1 (IL1) and the cytoplasmic C-terminal helix (CH) running parallel to the membrane (Supplementary Fig. 5). Superimposition of ovine $b^{0,+}AT$ with two structures of human $b^{0,+}AT$ recently reported[15,16] shows that our structures are in the same conformation as the previous ones. Notably, our cryo-EM map resolved numerous lipids, two of which were assigned to cholesterol and one to phospholipid (Fig. 2a,c; Supplementary Fig. 4d, e). Furthermore, the lipid nanodisc enclosing the TMD is bent by about ~30 degrees (Fig. 1c), which is reminiscent of the highly-curved membranes of the brush border microvilli, where system $b^{0,+}$ resides[20]. However, given that $b^{0,+}AT$–rBAT is also found in non-polarized cells where membrane curvature is presumably less, the lipid bending observed here could be specific to our sample preparation condition, and its physiological relevance needs further clarification.

**The $b^{0,+}AT$–rBAT interface.** Single-particle analysis indicated that the two $b^{0,+}AT$–rBAT subcomplexes are linked together with

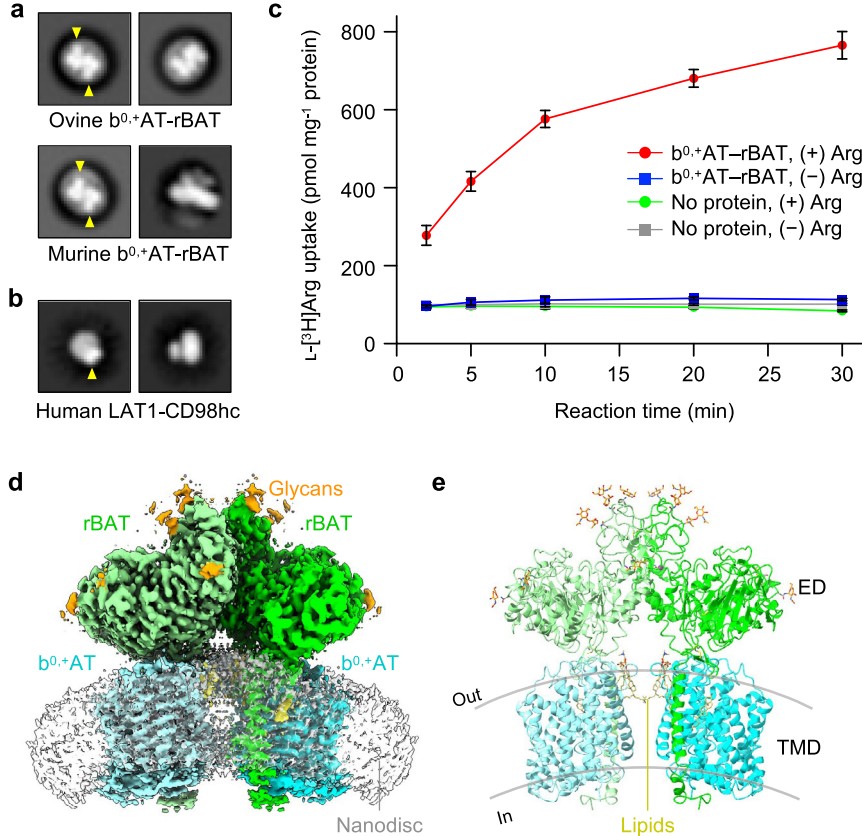

**Fig. 1 Structural and biochemical characterization of $b^{0,+}$AT–rBAT in a lipid environment. a** 2D class averages of ovine and murine $b^{0,+}$AT–rBAT imaged by negative-stain electron microscopy. The rBAT ectodomains are marked by yellow arrows. **b** 2D class averages of LAT1–CD98hc. The CD98hc ectodomain is marked by a yellow arrow. **c** Uptake of L-[$^3$H]Arg by ovine $b^{0,+}$AT–rBAT reconstituted into proteoliposomes. Proteoliposomes were pre-loaded with either 1 mM or no L-Arg. Control experiments were performed with protein-free liposomes. Values are mean ± SEM. $n = 3$ technical replicates. **d** Cryo-EM map of the super-dimeric $b^{0,+}$AT–rBAT complex in lipid nanodiscs. Map densities corresponding to different chains and chemical identities are colored as follows: rBAT (green and light green), $b^{0,+}$AT (cyan and light cyan), nanodisc (translucent white), lipids (yellow) and N-linked glycans (orange). **e** Ribbon diagram of $b^{0,+}$AT–rBAT. The position of the lipid bilayer is based on the lipid-solvent boundaries calculated by the PPM server (https://opm.phar.umich.edu/ppm_server).

substantial flexibility, which was also evident in the human complex that showed a blurred TMD map in the consensus 3D refinement[15,16]. To account for this flexibility, we performed multi-body refinement[17], taking individual heterodimers as rigid bodies moving relative to each other. This analysis not only revealed the swinging motions of $b^{0,+}$AT relative to the rigid rBAT core (Movie S1), but also yielded an improved map for the individual $b^{0,+}$AT–rBAT subcomplex at 3.0-Å nominal resolution (Supplementary Fig. 3c, f). This new map covered the intact $b^{0,+}$AT–rBAT heterodimer, which was not the case in the previous studies[15,16]. It therefore allowed us to analyze the $b^{0,+}$AT–rBAT interface in greater detail (Fig. 3).

On the extracellular side, EL2 of $b^{0,+}$AT is disulfide-bonded to the linker that connects TM1′ to the ectodomain of rBAT (Fig. 3b). EL4a and EL4b form salt bridges and hydrogen bonds to the Aα4-β5 loop and Aα6 (Fig. 3a). Notably, the Aα4-β5 loop is stabilized by Arg365′ (Fig. 3a), a residue associated with a prevalent cystinuria mutation R365W[1], suggesting that this substitution disrupts the $b^{0,+}$AT–rBAT extracellular interface. The extracellular interaction is further strengthened by the C-terminal peptide of rBAT (residues 651–685), which is absent in CD98hc (Fig. 3b)[9,10]. The first half of the peptide wraps around Cβ1–3 to form the ninth β-strand (Cβ9), which connects domain A and C (Fig. 3b). The second half folds into a cyclic β–hairpin, which interacts with $b^{0,+}$AT and touches the lipid bilayer

(Fig. 3b). Leu678′, which is the site of a known cystinuria mutation L678P[5], is located at the tip of this hairpin, suggesting a role in complex stabilization or membrane anchoring.

Within the membrane, TM1′ and TM4 form extensive helix-helix packing interactions (Fig. 3c). We identified one cholesterol molecule bound tightly at this interface (Fig. 2c; Supplementary Fig. 4a, d, e), which is in an equivalent position to that seen in LAT1–CD98hc[10], suggesting a conserved cholesterol-binding site in the HATs. Located near the cholesterol-binding site is Leu89′ (Supplementary Fig. 4a), which, when mutated to L89P, causes inefficient assembly with $b^{0,+}$AT, and cystinuria[12], highlighting its importance for heteromeric interactions. On the cytoplasmic side, the N-terminal helix (NH) of rBAT interacts with the C-terminal helix (CH) of $b^{0,+}$AT (Fig. 3d). The CH is unwound at Glu478. The subsequent residues extend into the cleft between NH and IL4 and are partially exposed to the cytoplasm (Fig. 3d). These residues correspond to the $^{480}$Val-Pro-Pro$^{482}$ motif, which has been suggested to interact with currently unknown cytoplasmic factors to regulate ER–Golgi trafficking[21]. Pro482, a residue frequently mutated in P482L that gives rise to a Japanese form of cystinuria[13], forms van-der-Waals interaction with IL4 (Fig. 3d; Supplementary Fig. 4b), highlighting its structural importance.

We previously noted that the heterodimerization interface might differ between HAT subgroups that associate with either rBAT or

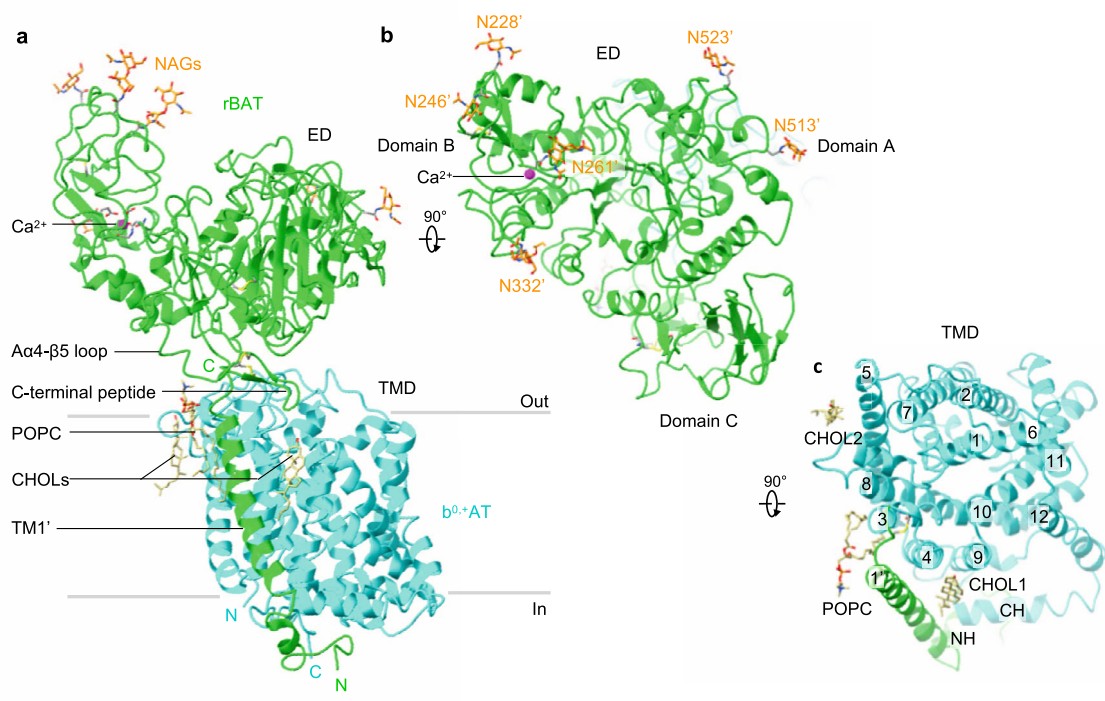

**Fig. 2 Structure of the ovine b^{0,+}AT–rBAT heterodimer.** Detailed depiction of the b^{0,+}AT–rBAT subcomplex, derived from 3D multi-body refinement (see Methods). Note that the complex is re-oriented from Fig. 1e to the membrane normal. CHOL, cholesterol; POPC, palmitoyl oleoyl phosphatidyl choline; TMD; transmembrane domain; ED, ectodomain; NAG, N-acetylglucosamine. **b** The rBAT ectodomain. The $Ca^{2+}$ ion, N-linked glycans, and individual subdomains are labeled. **c** b^{0,+}AT complexed with rBAT TM1′. Lipids and individual TMs are labeled. CH, C-terminal helix; NH, N-terminal helix.

CD98hc[10]. To investigate the difference, we superimposed the structure of b^{0,+}AT–rBAT onto LAT1–CD98hc using the TMDs, which showed that two ectodomains are displaced by ~40 Å to create distinct interfaces (Supplementary Fig. 6a). Moreover, surface electrostatic potential calculations show that the rBAT ectodomain is mostly negatively charged (Supplementary Fig. 6d), whereas CD98hc is positively charged[10], indicating different electrostatic interactions. In contrast to the distinct extracellular interactions, the transmembrane and cytoplasmic interactions are similar (Supplementary Fig. 6a). Given the low sequence conservation of TM1′, this observation is remarkable and could suggest strong evolutionary pressure acting on the intramembrane interactions of HATs.

**rBAT subdomains, disulfide bonds and glycosylation.** The domain B is unique to rBAT (not present in CD98hc) and composed of two insertion loops within domain A (Supplementary Fig. 5). For simplicity, we refer to the first loop as domain B-I (residues 213–289) and the second loop as domain B-II (residues 318–355) (Fig. 4a,b). Domain B-I has three β-strands and one α-helix, and contacts domain A through a cluster of hydrophobic residues (Fig. 4d). Thr216′ is located at the core of van-der-Waals interactions (Fig. 4d), suggesting how a common cystinuria mutation T216M[1] would destabilize the domain A–B interface. Likewise, the domain A–C interface has Met467′, which connects domains A and C through hydrophobic interactions (Fig. 4b, e), suggesting how M467T[1] would destabilize this interface. Domain B-II interacts with the adjacent rBAT molecule and thereby contributes to the formation of higher-order assemblies (Fig. 4b). The interaction involves six salt bridges, one π-cation-π stacking and several hydrophobic interactions (Fig. 4c), which together give rise to the strong higher-order assembly.

rBAT has three internal disulfide-bonds (Supplementary Fig. 5), which are formed during b^{0,+}AT-dependent oxidative folding[22,23]. The first of these is the functionally most important[22] and is formed between Cys242′ and Cys273′, stabilizing domain B (Supplementary Fig. 7a). The second bond is formed between Cys571′–Cys666′ in domain C, stabilizing the beta-sheet (Supplementary Fig. 7a). The third bond is formed between Cys673′ and the very C-terminus of rBAT, Cys685′, resulting in a cyclic C-terminal β-hairpin (Fig. 3b). Such a C-terminal disulfide bond is rarely seen in protein structures, highlighting the unique nature of the rBAT ectodomain.

Ovine rBAT has six N-linked glycans, three of which are shared in human (Supplementary Fig. 7b). Human rBAT has an additional glycan on Asn575′, which is critical for rBAT maturation[24]. Asn575′ in ovine rBAT is not glycosylated, due to the replacement of Ser577′ to Asn577′ (Supplementary Fig. 7b), which disrupts the third letter of the glycosylation motif NX(T/S). Structural comparison shows that the non-glycosylated Asn575′ of ovine rBAT adopts a typical β-sheet conformation, with its side-chain interacting with domain A (Supplementary Fig. 7c). By contrast, the glycosylated Asn575′ of human rBAT is flipped out of the β-sheet and is exposed to the solvent, with its glycan moiety interacting with domain A (Supplementary Fig. 7c). This structural difference is caused by the bulky amino acids Tyr425′ and Tyr579′ pushing away Asn575′ in human rBAT, which are replaced by smaller amino acids Ser425′ and Ser579′ in ovine rBAT. Thus, human rBAT has an additional glycan for stabilizing the domain A–C interface, which explains why disrupting this glycan causes premature degradation of system b^{0,+} in humans[24].

**$Ca^{2+}$-binding site.** The cryo-EM map of rBAT revealed a strong spherical density in the middle of domain B (Fig. 5a). This density

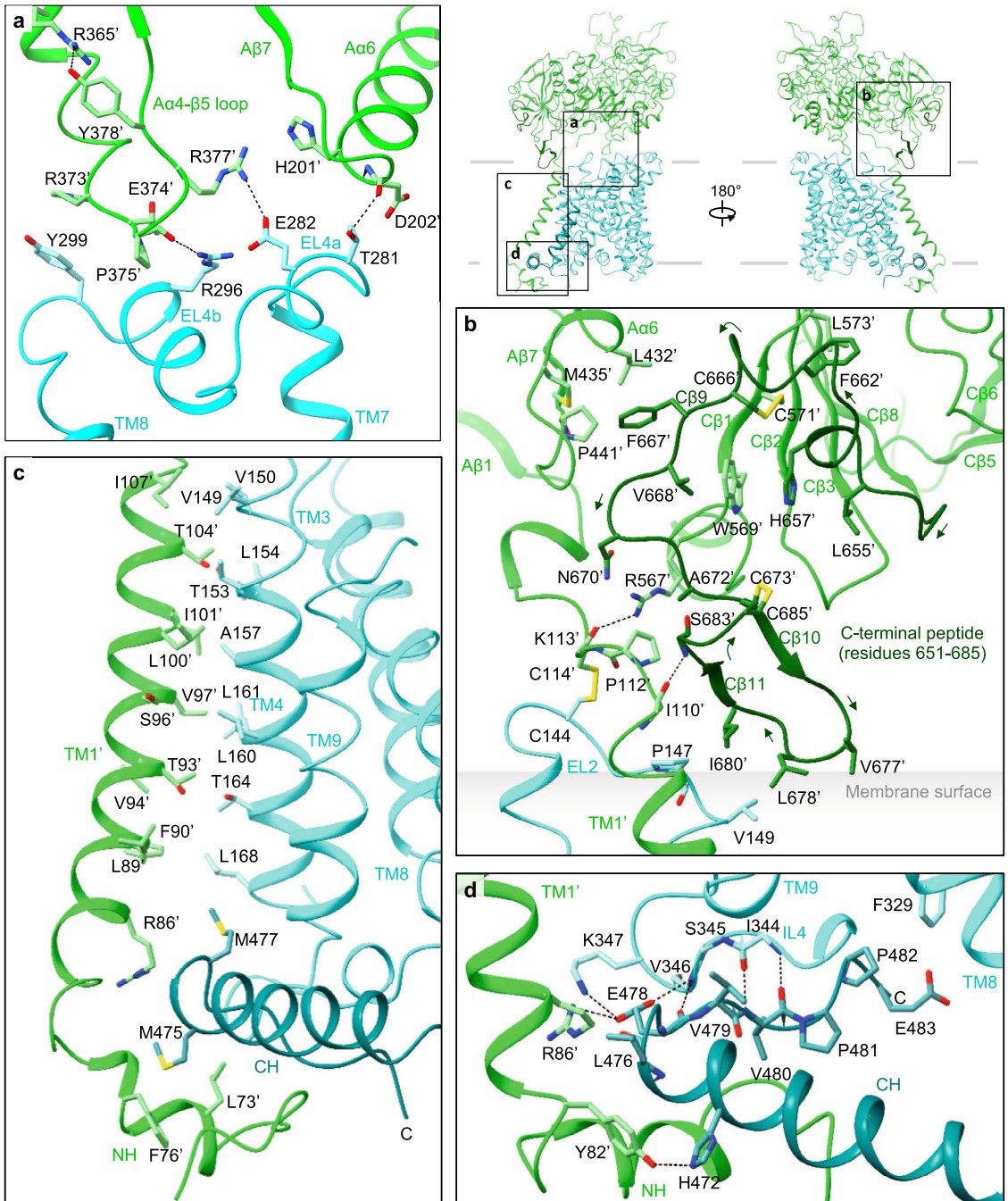

**Fig. 3 The interface analysis of b$^{0,+}$AT–rBAT. a** Extracellular interactions formed by the Aα4-β5 loop, Aα6, EL4a and EL4b. Key interacting residues are shown, with hydrophilic interactions depicted by dotted lines. Arg365′ stabilizes the Aα4-β5 loop through interaction with Tyr378′. The locations of panels (**a–d**) are indicated on the right. **b** C-terminal peptide of rBAT. The peptide is dark green for better visibility. Leu678′ contacts the membrane surface, as calculated by the PPM server (https://opm.phar.umich.edu/ppm_server). **c** Intramembrane interactions. TM1′ and TM4 have numerous hydrophobic residues that pack against each other. The C-terminal helix (CH) of b$^{0,+}$AT is dark cyan for better visibility. **d** CH and 'Val-Pro-Pro' motif of b$^{0,+}$AT, forming intracellular interactions.

is surrounded by the acidic side chains of Asn214′, Asp284′ and Glu321′ and the main chain carbonyl groups of Tyr318′ and Leu319′, indicative of a metal cation (Fig. 5b). Based on the coordination geometry and the sequence homology with known Ca$^{2+}$-binding sites of amylases[25], we assigned this density to a bound Ca$^{2+}$ ion. As we did not supply any Ca$^{2+}$ during purification, the bound ion has most likely been acquired during protein expression. This density was also observed in the recent structures of human rBAT and assigned as Ca$^{2+}$, supporting a conserved binding site[15,16].

To gain insight into the Ca$^{2+}$-binding site, we compared the rBAT ectodomain with homologous glucosidases with conserved Ca$^{2+}$-binding sites. A homology search revealed close structural similarity to an α-amylase from *Anoxybacillus* species, known as TASKA[25], and indicated that the observed Ca$^{2+}$ site in rBAT corresponds to 'site 1' of TASKA, which stabilizes the protruding loop of domain B[25]. In rBAT, the Ca$^{2+}$ ion bridges domains B-I and B-II to reinforce the super-dimer interface (Fig. 4c,d). Electrostatic potential calculations of rBAT with or without Ca$^{2+}$ showed that Ca$^{2+}$ neutralizes the super-dimer

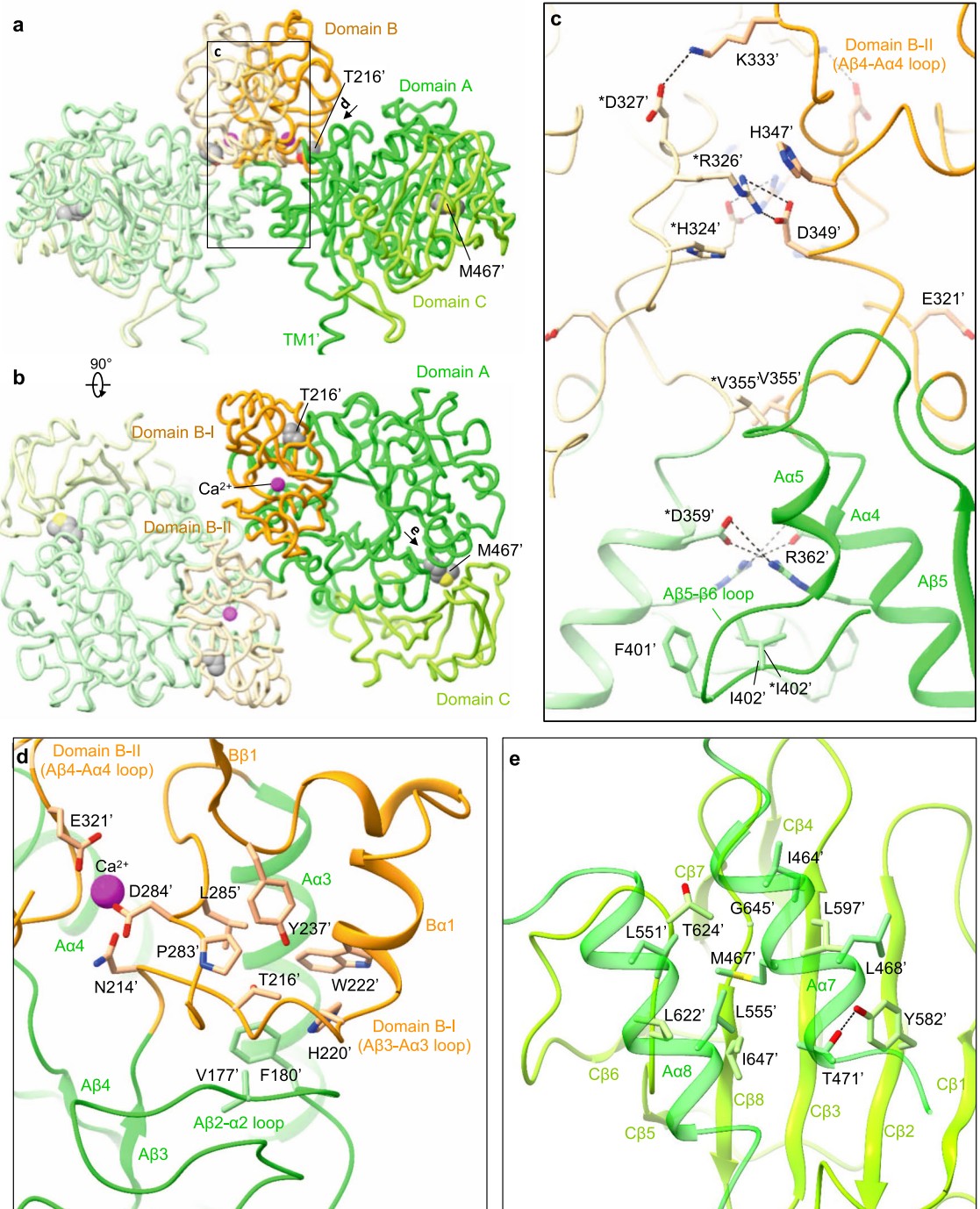

**Fig. 4 The rBAT ectodomain. a** Structure of the ectodomain homo-dimer. The three subdomains are A (green), B (orange) and C (light green). Common cystinuria-related residues Thr216′ and Met467′ and the $Ca^{2+}$ ions are shown as spheres. **b** Extracellular view of the rBAT ectodomain. **c** Zoom-up of super-dimer interface. Salt bridges, π-cation-π stacking and van-der-Waals interactions are indicated. Residues from the adjacent protomer are marked with asterisks (*). **d** Interaction network around Thr216′ and $Ca^{2+}$. **e** Interaction network around Met467′.

interface to facilitate the homomeric interaction (Supplementary Fig. 8a, b). Although sequence homology suggested the presence of another $Ca^{2+}$-binding site, known as 'site 2' in TASKA (Supplementary Fig. 7d), this site is occupied by a weaker peak in our cryo-EM map, indicating no or sub-stoichiometric binding (Supplementary Fig. 7e). Although $Ca^{2+}$ ions regulate the enzymatic activity of amylases[26,27], there is no evidence for any enzymatic activity of rBAT. Therefore, we explored a different physiological role of $Ca^{2+}$.

**The role of $Ca^{2+}$ in super-dimerization, maturation and function of system $b^{0,+}$.** Many ER-resident chaperones and post-translational modification machineries, such as calreticulin[28,29], BiP/GRP78[30,31] and the protein disulfide isomerase[32], are known to bind $Ca^{2+}$ as a regulatory element[33]. Therefore, we hypothesized that the bound $Ca^{2+}$ in rBAT may be involved in the system $b^{0,+}$ biogenesis. To test this hypothesis, we investigated the N-glycan maturation of system $b^{0,+}$ using the Endo H assay previously established[12]. In this assay, fully mature glycosylated

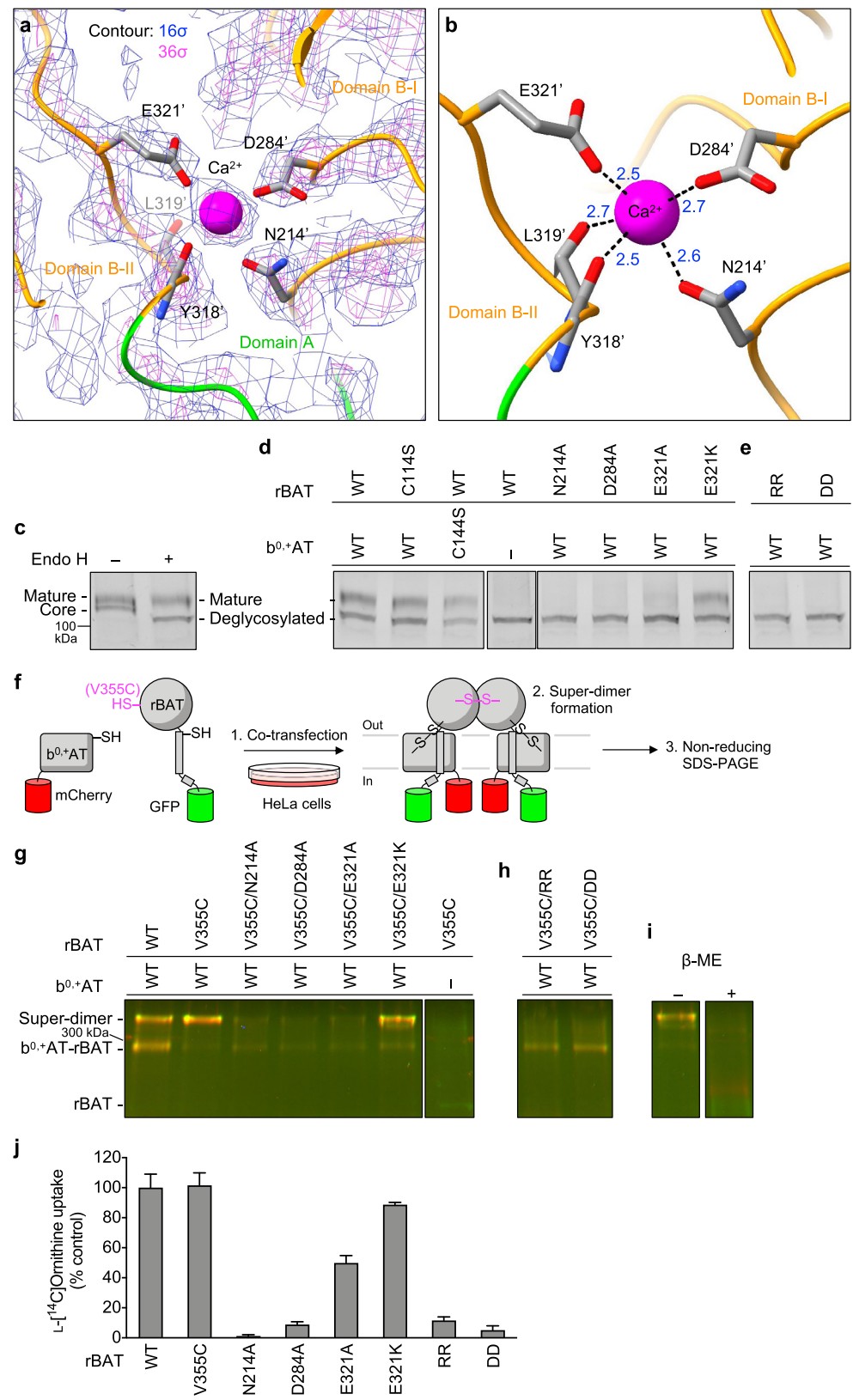

proteins that reached the Golgi apparatus are resistant to Endo H treatment, whereas core glycosylated ones that are trapped in the ER remain Endo H sensitive, thereby allowing us to investigate protein maturation as a proxy for protein trafficking. We modified the assay by fusing GFP to rBAT and mCherry to $b^{0,+}AT$, to enable fluorescence-based band detection in SDS-PAGE

(Supplementary Fig. 9a,b). As a positive control, co-expression of fluorescence-tagged wild-type rBAT and $b^{0,+}AT$ yielded an Endo H-resistant rBAT band (Fig. 5c). By contrast, rBAT alone did not mature in the absence of $b^{0,+}AT$ (Fig. 5d: lane 4), confirming the requirement of $b^{0,+}AT$ for rBAT maturation[12]. Using this assay, we investigated the effects of the $Ca^{2+}$-binding site

**Fig. 5 The Ca²⁺-binding site. a** Cryo-EM map of the Ca²⁺-binding site at 2.6 Å resolution, contoured at 16 σ (blue) and 36 σ (magenta). **b** Close-up view of the Ca²⁺-binding site. The coordination distances in Å are labeled. Note that unmodelled water molecules may contribute to the full Ca²⁺ coordination. **c** Endo H assay evaluating the N-glycan maturation. The wild-type rBAT yields a mature, higher molecular-weight form that resists Endo H treatment (upper band), in addition to the core-glycosylated form (lower band, left lane), which gets de-glycosylated by the Endo H treatment (lowest band, right lane). **d** Endo H assay for the Ca²⁺-binding site. Lanes 1–3: positive controls, where WT or disulfide-less mutants (rBAT C114S or b⁰,⁺AT C144S) showed normal rBAT maturation. Lane 4: negative control, where rBAT without b⁰,⁺AT lost the mature band. Lanes 5–8: Ca²⁺-binding site mutants. Cation-compensating E321K restored the mature band nearly to the WT level. **e** Endo H assay for super-dimerization mutants (locations shown in Fig. 4c). The two double mutations D349R/D359R (labeled RR) and R326D/R362D (labeled DD) completely abolished maturation. **f** Workflow of site-specific cross-linking assay for detecting super-dimers. V355C introduces a pair of cysteines at the rBAT–rBAT homomeric interface (locations shown in Fig. 4c). When b⁰,⁺AT–rBAT forms a super-dimer, a pair of Cys355′ residues form a disulfide bond, which can be detected as higher-molecular weight species in oxidizing SDS-PAGE. Also see Supplementary Fig. 10 for larger gels. **g** Cross-linking assay for Ca²⁺-binding site mutants. Overlay displays two fluorescence channels (GFP (green) and mCherry (red)) on non-reducing SDS-PAGE. Yellow thus represents the b⁰,⁺AT–rBAT heterocomplexes (monomeric or super-dimeric). Note that wild-type rBAT mutations show some super-dimers even without the V355C mutation, indicating a strong assembly. **h** Cross-linking assay for super-dimer interface mutants. **i** Control experiments for cross-linking assay. Addition of β-mercaptoethanol dissociates b⁰,⁺AT and rBAT into two separate bands. **j** Uptake of 100 μM L-[¹⁴C]-ornithine in HEK293 cells transfected with b⁰,⁺AT-WT, and rBAT-WT or rBAT mutants. Net uptake was normalized to WT set as 100%. Values are mean ± SEM. n = 4 technical replicates.

mutations, N214A, D284A and E321A, on N-glycan maturation. Intriguingly, all three mutants showed significantly decreased maturation (Fig. 5d: lanes 5–7), demonstrating that Ca²⁺ binding is important for protein maturation. Given that Ca²⁺ introduces a positive charge to stabilize the cluster of negatively-charged residues, we next asked if adding a positive charge can complement these mutants. To this end, we prepared an E321K mutant, which would mimic the positive charge of Ca²⁺ to compensate for its loss. Indeed, the E321K mutant restored the maturation (Fig. 5d: lane 8) confirming the importance of a positive charge at this position for rBAT maturation. Notably, some amylases naturally have Lys at this position (e.g., Lys206 in a GH13 amylase; PDB 5ZCC)[34], suggesting a conserved yet substitutable role of Ca²⁺.

Given that Ca²⁺ and domain B are both structurally involved in super-dimer formation (Fig. 4c, d), we reasoned that those mutations in the Ca²⁺-binding site first affect the super-dimerization and thereby impair protein maturation. To test this assumption, we evaluated super-dimer formation of the mutants using a site-specific disulfide cross-linking assay (Fig. 5f–i; Supplementary Fig. 10). As a site for introducing cysteine, we chose Val355′, which interacts with *Val355′ of the adjacent protomer, with a Cα–Cα distance of ~4.4 Å (Fig. 4c), suitable for spontaneous disulfide-bond formation. Indeed, when co-expressed with b⁰,⁺AT in HeLa cells, rBAT V355C showed the cross-linked super-dimer for nearly 100% under non-reducing SDS-PAGE (Fig. 5g: lanes 1–2), confirming site-specific cross-linking. This cross-linking did not occur without b⁰,⁺AT, indicating that the rBAT–rBAT association requires b⁰,⁺AT (Fig. 5g: lane 7). To investigate the effect of Ca²⁺ on super-dimer formation, we introduced the mutants N214A, D284A or E321A into this V355C background. In all of them, super-dimer formation was reduced (Fig. 5g: lanes 3–5), revealing the importance of Ca²⁺ for super-dimer formation. The cation-compensating mutation, E321K, restored the super-dimer (Fig. 5g: lane 6), supporting a role of the positive charge for the super-dimer formation.

To analyze the importance of Ca²⁺ for protein function, we next performed amino acid transport assays in HEK293 cells transfected with b⁰,⁺AT and rBAT without any fluorescent tags. Prior to the analysis, we confirmed that fluorescence tags or host cell types did not significantly alter protein function or complex formation (Supplementary Fig. 11). Three Ca²⁺-binding site mutants, N214A, D284A and E321A, showed significantly reduced Orn uptake (Fig. 5j), indicating cellular effects of these mutants. By contrast, the cation-compensating mutant E321K restored the activity close to 100%, confirming the functional role

of positive charge (Fig. 5j). Furthermore, the V355C mutant, which was used for the cross-linking assay, showed negligible effect on transport activity, validating our assay. These results support our hypothesis that Ca²⁺ and super-dimerization are key factors for system b⁰,⁺ maturation and function.

To confirm the importance of super-dimerization per se, we directly mutated the residues at the super-dimer interface. Two double mutants, D349R/D359R and R326D/R362D, disrupt two critical salt bridges at the rBAT–rBAT interface (Fig. 4c) and would thus abolish the super-dimer. Indeed, when tested with cross-linking assay, both V355C/D349R/D359R and V355C/R326D/R362D did not form super-dimers while retaining the heterodimeric assembly, confirming our mutant designs (Fig. 5h). We thus tested these mutants by the Endo H assay, which showed impaired N-glycan maturation (Fig. 5e). Accordingly, these mutants showed lower transport activities (Fig. 5j). Altogether, these results revealed an unexpected role of Ca²⁺ for the formation of higher-order assemblies, maturation, and cellular function of system b⁰,⁺.

**ER Ca²⁺ mediates super-dimerization for system b⁰,⁺ biogenesis.** Model of b⁰,⁺AT–rBAT biogenesis was previously proposed to include multiple steps, starting from the initial assembly of b⁰,⁺AT with core-glycosylated rBAT (rBAT_c), followed by oxidative folding of the rBAT ectodomain in the ER and N-glycan maturation[12,22,35]. We hypothesized that Ca²⁺ binding to domain B is another key step, facilitating super-dimer formation and its subsequent trafficking out of the ER (Fig. 6a). To clarify this, we performed pulse-chase assays for wild-type b⁰,⁺AT, and wild-type rBAT or its Ca²⁺-binding site mutants in HeLa cells, building on previous experiments that established the biogenesis models of system b⁰,⁺[12,24]. At 0–0.5 h chase time, super-dimers from wild-type rBAT were hardly detected, while ladders of rBAT_c monomers and b⁰,⁺AT–rBAT_c heterodimers were already synthesized at significant amounts and getting saturated (Fig. 6b: lanes 1–2), confirming the assembly of b⁰,⁺AT and rBAT_c as the initial step in biogenesis. At 1 h, rBAT monomers remained only non- or low-glycosylated forms, suggesting the glycosylated ones turned into super-dimers or mature b⁰,⁺AT-rBAT heterodimers as shown in the increase of their bands (Fig. 6b: lane 3). At the later chase time (2–4 h), the amounts of super-dimers and mature b⁰,⁺AT-rBAT heterodimers increased and gradually saturated (Fig. 6b: lanes 4–6), supporting the model in which super-dimerization occurs as the latter step after an initial assembly of b⁰,⁺AT and rBAT_c. The decline in rBAT_c monomers was likely the mixed results of super-dimer formation as shown here and partial degradation as demonstrated previously[12].

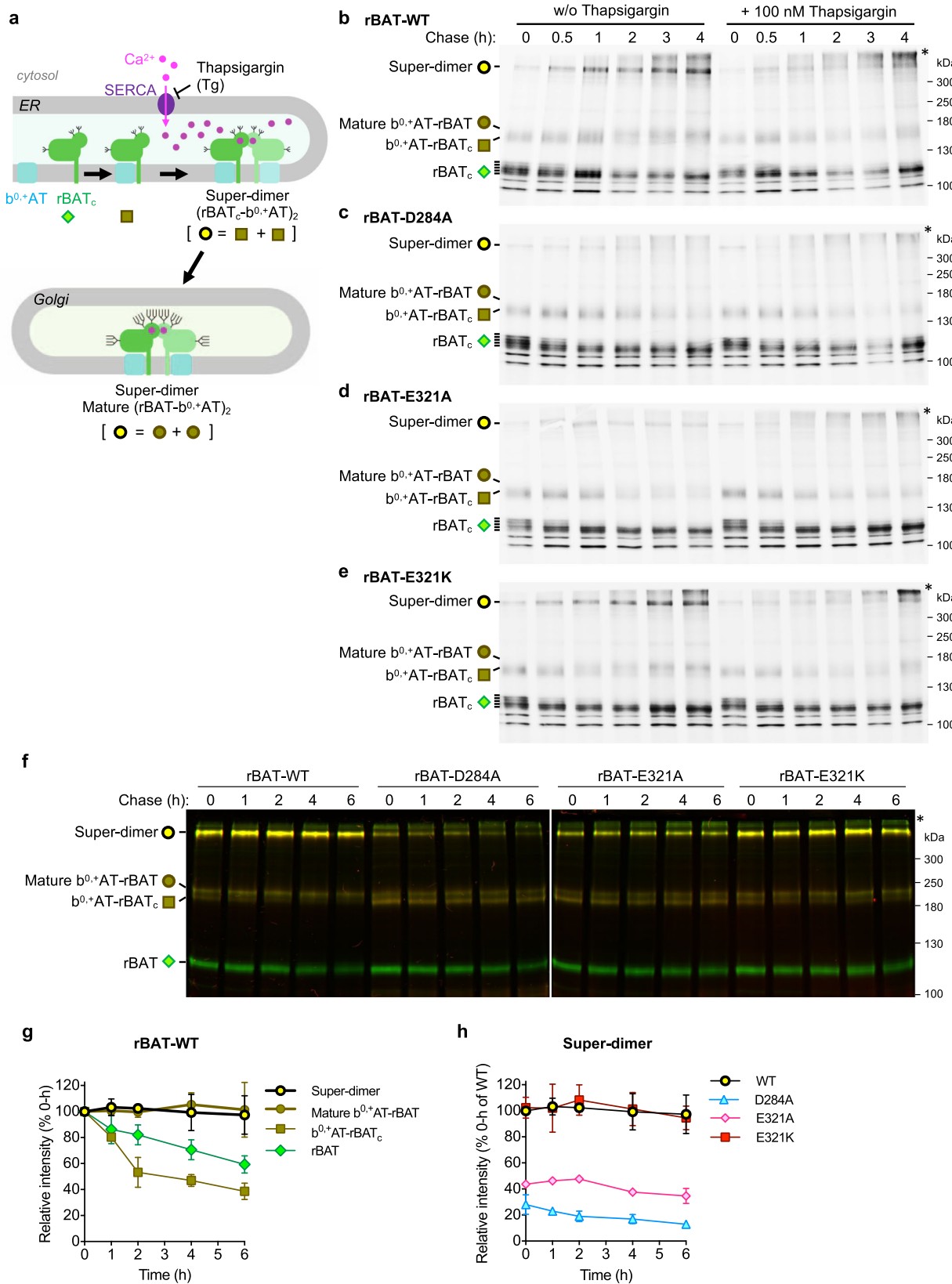

Using the same pulse-chase assay, we analyzed the role of ER $Ca^{2+}$ by applying sarco endoplasmic reticulum $Ca^{2+}$ ATPase pump (SERCA) inhibitor thapsigargin (Tg) to inhibit the $Ca^{2+}$ flux into the ER lumen[36]. The application of Thapsigargin did not significantly alter the amounts of $rBAT_c$ monomers and $b^{0,+}AT$–$rBAT_c$ heterodimers, but markedly diminished super-dimers and mature $b^{0,+}AT$-rBAT heterodimers (Fig. 6b: lanes 10–12; Supplementary Fig. 12a). At the late time and high concentration of Tg, the aggregations were observed at the top of the gels, suggesting protein instability and degradation from the failure of super-dimerization (Fig. 6b: lanes 11–12; Supplementary Fig. 12a). The effective concentration of Tg for inhibiting

**Fig. 6 $b^{0,+}$AT-rBAT biogenesis and protein stability. a** Schematic model proposing $b^{0,+}$AT-rBAT biogenesis. The process is initiated by dimerization of $b^{0,+}$AT and rBAT$_c$ followed by their super-dimerization, both of which take place in the ER. Super-dimer is then translocated to the Golgi apparatus for N-glycan maturation. Symbols below each protein form correspond to those presented on the gels in **b–h**. **b–e** Pulse-chase analysis of HeLa cells transfected with wild-type $b^{0,+}$AT (no tag) and wild-type GFP-rBAT **b** or the Ca$^{2+}$-binding site mutants (D284A, E321A, or E321K; **c–e**). After labeling the cells with [$^{35}$S]Met/Cys, the cells were chased at indicated times with or without 100 nM thapsigargin. The eluent from immunoprecipitation were subjected to 7% non-reducing SDS-PAGE. Three major forms of rBAT were detected during chasing times: super-dimer, $b^{0,+}$AT-rBAT dimers (mature and immature forms), and core glycosylated rBAT (rBAT$_c$) monomer. Asterisk indicates the aggregation of rBAT at the top of polyacrylamide gel. **f** Cycloheximide (CHX)-chase analysis of HeLa cells transfected with wild-type mCherry-$b^{0,+}$AT and wild-type GFP-rBAT or the Ca$^{2+}$-binding site mutants (D284A, E321A, or E321K). After transfection for 16 hrs, the cells were treated with 50 μg/ml CHX at indicated times. Equal amount of total protein lysates were subjected to 7% non-reducing SDS-PAGE. Two gel images display the overlayed GFP (green) and mCherry (red) fluorescence channels imaged at the same exposure time (30 s for GFP and 5 min for mCherry). Yellow thus represents the merge of mCherry-$b^{0,+}$AT and GFP-rBAT. **g** Band intensity analysis of the wild-type $b^{0,+}$AT-rBAT. The intensities of rBAT in all complex forms (super-dimers, mature $b^{0,+}$AT-rBAT dimers, $b^{0,+}$AT-rBAT$_c$ dimers, and rBAT monomer) were quantitated and normalized to their values at 0 h. Graphs are mean ± SD. $n = 3$ (independent experiments including the representative gels in **f**). **h** Band intensity analysis of the super-dimers from the wild-type $b^{0,+}$AT-rBAT and the Ca$^{2+}$-binding site mutants. The band intensities of super-dimers from rBAT mutants were compared to that of wild-type rBAT at 0 h. Graphs are mean ± SD. $n = 3$ (independent experiments including the representative gels in **f**).

super-dimerization was as low as 10 nM (Supplementary Fig. 12a), consistent with its effective concentration for SERCA inhibition[36–38]. We also tested whether a Ca$^{2+}$-chelator EGTA can strip off Ca$^{2+}$ to dissociate the super-dimer, but the addition of 10 mM EGTA to solubilized $b^{0,+}$AT–rBAT had no effect on the protein size or stability, suggesting a strong and irreversible binding of Ca$^{2+}$ (Supplementary Fig. 13a). These results support that Ca$^{2+}$ is essential in the assembly of system $b^{0,+}$ and is acquired in the ER.

Next, we tested the effect of the Ca$^{2+}$-binding site mutations on protein biogenesis by the pulse-chase assay. The Ca$^{2+}$-binding site mutants (D284A and E321A) revealed a profile of rBAT$_c$ monomer similar to the wild-type but hardly produced super-dimers and mature $b^{0,+}$AT-rBAT heterodimers, supporting the indispensable role of Ca$^{2+}$ in super-dimerization (Fig. 6c, d: lanes 1–6). In addition, Tg completely abolished super-dimerization of these mutants (Fig. 6c, d: lanes 9–12; Supplementary Fig. 12b–d). The cation-compensating E321K restored the wild-type behavior (Fig. 6e: lanes 1–6; Supplementary Fig. 12e). Notably, super-dimers in E321K were still sensitive to Tg, as in the wild-type (Fig. 6e: lanes 9–12). We speculate that Tg also limits the functions of known Ca$^{2+}$-regulated ER resident proteins which may be critically involved in super-dimer formation. Nevertheless, such effects were not observed in wild-type LAT1-CD98hc (Supplementary Fig. 12f), which neither has a Ca$^{2+}$-binding site nor forms a super-dimer[9,10], supporting the specific role of the ER Ca$^{2+}$ in system $b^{0,+}$.

To study the stability of super-dimers within cells, we next performed a cycloheximide (CHX)-chase assay and monitored the behaviors of different biogenesis intermediates of system $b^{0,+}$ (Fig. 6a). The results showed that super-dimers and mature $b^{0,+}$AT–rBAT heterodimers were relatively stable (Fig. 6f, g). In contrast, $b^{0,+}$AT–rBAT$_c$ and monomeric rBAT continuously decreased since the beginning of the chase time (Fig. 6f, g), consistent with our pulse-chase experiments (Fig. 6b–e) and the previous study that revealed fast degradation of $b^{0,+}$AT–rBAT$_c$ and rBAT monomer in the ER[12]. D284A and E321A mutants displayed faint super-dimers, while E321K recovered it to similar amounts as for the wild-type (Fig. 6f, h). Analysis of the proteins solubilized in detergent using FSEC showed that only wild-type or E321K yields a monodisperse peak, while other mutants (N214A, D284A, and E321A) were unstable upon extraction by detergent and appear as high-molecular weight aggregates (Supplementary Fig. 13b). These results show that Ca$^{2+}$ binding and super-dimerization assist protein stability within the cells.

To validate the impacts of mutations to protein trafficking, we analyzed sub-cellular localization of the Ca$^{2+}$-binding site mutants by staining the transfected cells with an ER-marker PDI (Fig. 7a) or a Golgi-marker RCAS1 (Fig. 7b), and correlating those signals with fluorescence signals of GFP-rBAT and $b^{0,+}$AT-mCherry. The results showed that wild-type $b^{0,+}$AT–rBAT were mainly localized at the plasma membrane and partially at the Golgi apparatus, whereas the N214A, D284A and E321A mutants were mostly trapped in the ER (Fig. 7a; Supplementary Fig. 14), confirming the importance of Ca$^{2+}$ at early stages of protein trafficking. We note that a minor fraction of cells transfected with E321A showed plasma membrane localization, which coincides with its partially retained transport activity (Fig. 5j), suggesting that E321A is less severe than N214A and D284A. The cation-compensating mutant E321K restored protein localization at the plasma membrane (Fig. 7a, b), verifying its compensation property. Similar fluorescence imaging for the corresponding mutants on a V355C mutant background showed the same localization patterns, validating the cross-linking assay (Supplementary Fig. 14). Taken together, these results show that Ca$^{2+}$ binding in the ER is the key step for super-dimer formation and its trafficking to the plasma membrane.

**rBAT domain B and cystinuria.** Given the unexpected observation that Ca$^{2+}$ binding to domain B is an important requirement for $b^{0,+}$AT–rBAT biogenesis and trafficking, we asked if any of the known cystinuria mutations exhibit correlated phenotypes to the Ca$^{2+}$-binding site mutants. To test this, we investigated representative cystinuria mutants from each subdomain, namely L89P (TM1'), T216M (domain B), R365W (the domain A-TMD interface), M467T (the domain A–C interface), L678P (domain C), Δ673-685 (C-terminal deletion) and $b^{0,+}$AT G105R (TMD)[1,39]. Consistent with the previous studies[12,40], rBAT L89P diminished the amount of $b^{0,+}$AT–rBAT heterodimer and thus also the super-dimer (Fig. 8a: lane 3), accompanied by the low level of N-glycan maturation (Fig. 8b: lane 2), supporting that L89P mutant impairs the heteromeric interaction between $b^{0,+}$AT and rBAT and thereby decrease the functional complexes. $b^{0,+}$AT G105R showed almost no expression (Supplementary Fig. 9b), causing the loss of rBAT maturation (Fig. 8b: lanes 2-3), consistent with previous observation that $b^{0,+}$AT is required for oxidative folding and maturation of the rBAT ectodomain[22]. Fluorescent imaging confirmed that these two mutants are trapped in the ER (Fig. 8c), and cell-based transport assays showed that they have no transport function (Fig. 8d). Interestingly, the mutant T216M in domain B behaved similarly to the Ca$^{2+}$-binding mutants, as it retained the $b^{0,+}$AT-rBAT heterodimer but completely lost the super-dimer (Fig. 8a: lane 4). Correspondingly, T216M proteins were predominantly localized in the ER (Fig. 8c), and they exhibited disrupted N-glycan

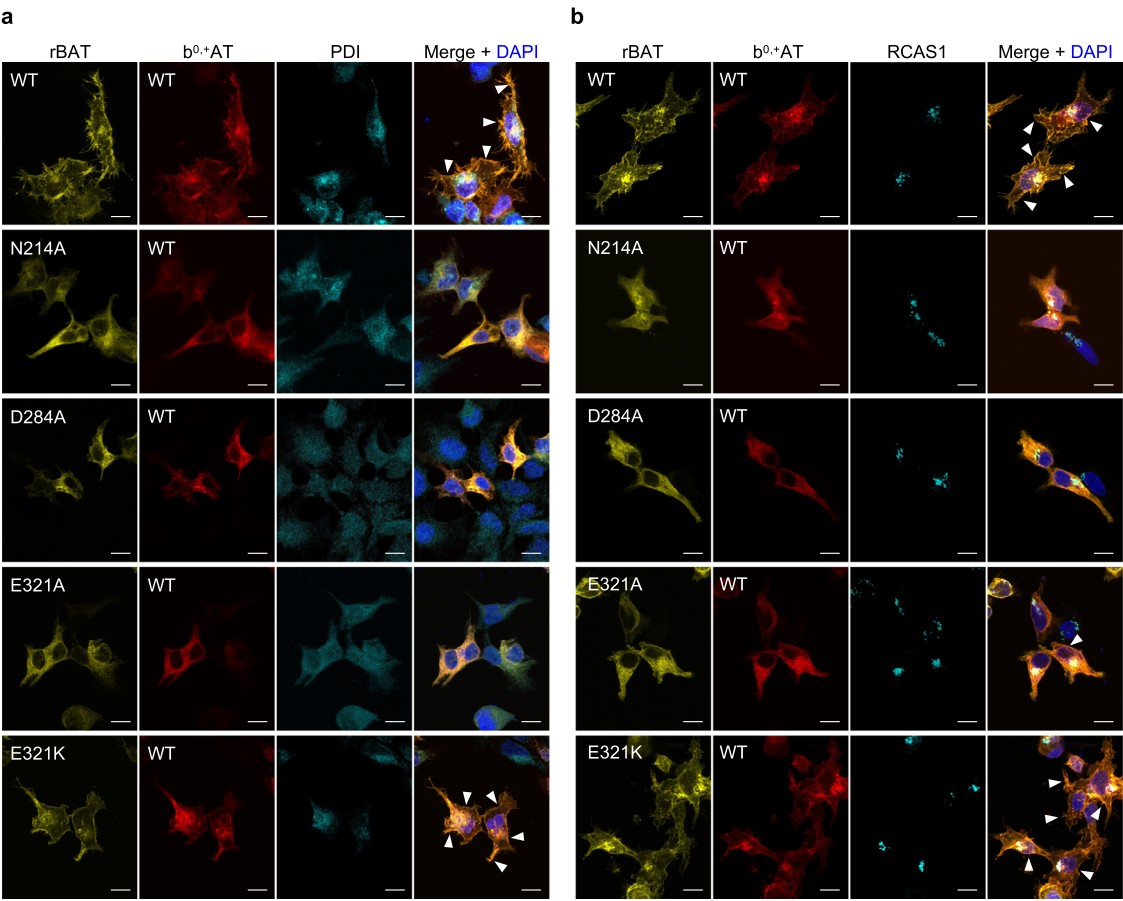

**Fig. 7 Localization of $b^{0,+}$AT-rBAT and its $Ca^{2+}$-binding site mutants.** Fluorescence imaging of HeLa cells expressing wild-type $b^{0,+}$AT-mCherry (red) and wild-type or mutated GFP-rBAT (yellow). Fluorescent staining with anti-PDI (panel **a**, colored cyan) and anti-RCAS1 (panel **b**, colored cyan) antibodies were used as markers for the ER and Golgi apparatus, respectively. Plasma membrane localization of $b^{0,+}$AT-rBAT are indicated in the white arrowheads. Scale bar = 10 μm.

maturation (Fig. 8b: lane 4) and low transport function (Fig. 8d). These results suggest that the T216M mutation in domain B disrupts the higher-order assembly and prevents the N-glycan maturation, and thereby leads to cystinuria. The mutants for other subdomains, namely R365W, M467T and Δ673-685, could form super-dimers to some extent (Fig. 8: lanes 5-6,8), while showing decreased N-glycan maturation and transport functions (Fig. 8b, d), consistent with a previous study[22] and supporting their contribution to the rBAT ectodomain folding and stability. L678P in domain C did not significantly alter maturation (Fig. 8a: lane 7) or super-dimerization (Fig. 8b: lane 7). L678P may affect the protein stability or membrane anchoring since the C-terminal peptide of rBAT mediates interactions with $b^{0,+}$AT and the lipid bilayer (Fig. 3b).

**Substrate specificity of $b^{0,+}$AT.** Our $b^{0,+}$AT structure adopts the inward-facing conformation without any substrate bound. As in other LeuT-fold transporters, a putative substrate-binding site of $b^{0,+}$AT is formed between the hash and bundle domains[41,42] (Fig. 9a, b). At this site, the two broken helices TM1 and TM6 expose the main-chain amino and carbonyl groups to the cytoplasmic solvent (Fig. 9c, d), forming a substrate backbone recognition site. To gain insight into possible substrate recognition mechanisms, we compared our $b^{0,+}$AT structure with GkApcT[43], a bacterial homolog of the cationic amino acid transporters (CATs; SLC7A1–4)[44]. In the Arg-bound structure of GkApcT (M321S)[43], the substrate backbone is recognized by the main chains of TM1 and TM6 and the substrate guanidium group is recognized by Glu115 on TM8 (Fig. 9e). This

Glu115 is not conserved in $b^{0,+}$AT, but instead Asp233 would be located adjacent to the guanidium group (Fig. 9e), suggesting its involvement in the positive charge recognition. GkApcT has another acidic residue Asp237, which is replaced by a neutral residue Asn236 in $b^{0,+}$AT, suggesting that one acidic residue is sufficient for recognizing cationic substrates in system $b^{0,+}$.

Because $b^{0,+}$AT also transports neutral amino acids, we compared $b^{0,+}$AT with LAT1 (SLC7A5), a large neutral amino acid transporter of the SLC7 family. The key acidic residue Asp233 of $b^{0,+}$AT corresponds to Gly255 of LAT1 (Fig. 9f), which was shown to be important for recognizing large amino acids[10]. With this small Gly255 residue, LAT1 forms a so-called distal pocket, which might accommodate large hydrophobic substrates. In $b^{0,+}$AT, this distal pocket is disrupted (Fig. 9f) by the replacement of Val408 with a larger Tyr408, along with Asp233. Therefore, $b^{0,+}$AT would not prefer bulky amino acids, unlike LAT1, which can recognize much bulkier analogs such as $T_3$ and JPH203[45].

Multiple sequence alignment of HATs, CATs[44], and GkApcT showed that the unwound regions of TM1 (residues 44–48 in $b^{0,+}$AT) are highly conserved (Supplementary Fig. 15a), whereas the equivalent region of TM6 (residues 233–237 in $b^{0,+}$AT) varies substantially (Supplementary Fig. 15b). Asp233 and Asn236 in $b^{0,+}$AT have drawn our attention, since at least one of these residues is negatively charged in systems $b^{0,+}$, $y^+$L, and CATs, which all can transport cationic amino acids (Supplementary Fig. 15b). To test the importance of these residues for substrate specificity, we mutated Asp233 and Asn236 and measured transport

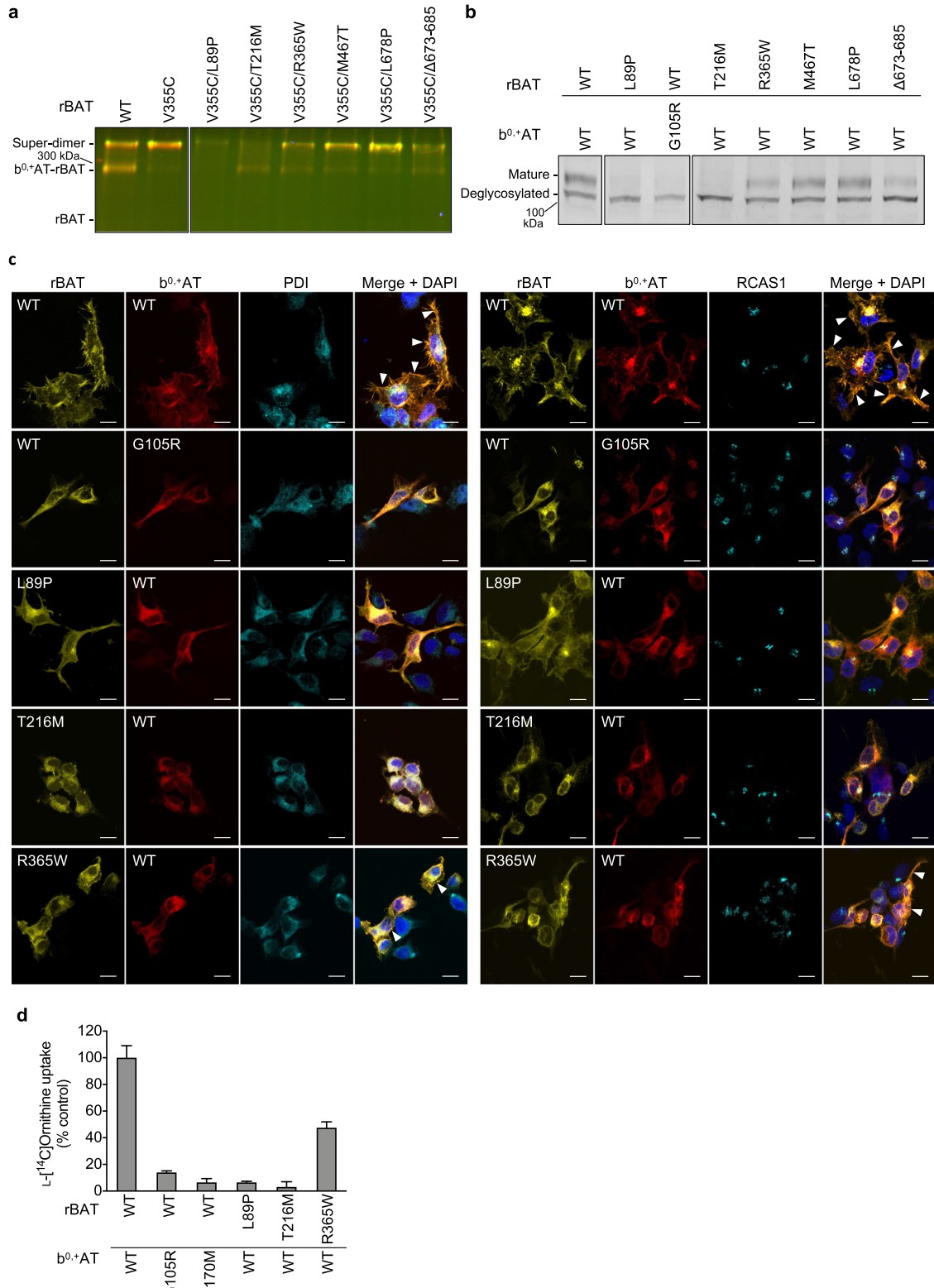

**Fig. 8 Cystinuria mutants. a** Cross-linking assay to evaluate the super-dimerization for cystinuria mutants. Overlay of GFP and mCherry fluorescence channels on non-reducing SDS-PAGE are displayed. **b** Endo H assay to evaluate the N-glycan maturation for cystinuria mutants. **c** Fluorescence imaging of HeLa cells expressing GFP-rBAT (yellow) and mCherry-$b^{0,+}$AT (red) or the indicated mutants. Fluorescent staining with anti-PDI (cyan in the left panel) and anti-RCAS1 (cyan in the right panel) antibodies were used as markers for ER and Golgi apparatus, respectively. Plasma membrane localization of $b^{0,+}$AT-rBAT are indicated in the white arrowheads. Scale bar = 10 μm. **d** Uptake of 100 μM L-[$^{14}$C]-ornithine in HEK293 cells transfected with wild-type $b^{0,+}$AT and rBAT or the indicated mutants. Net uptake was normalized to WT set as 100%. Values are mean ± SEM. $n = 4$ technical replicates.

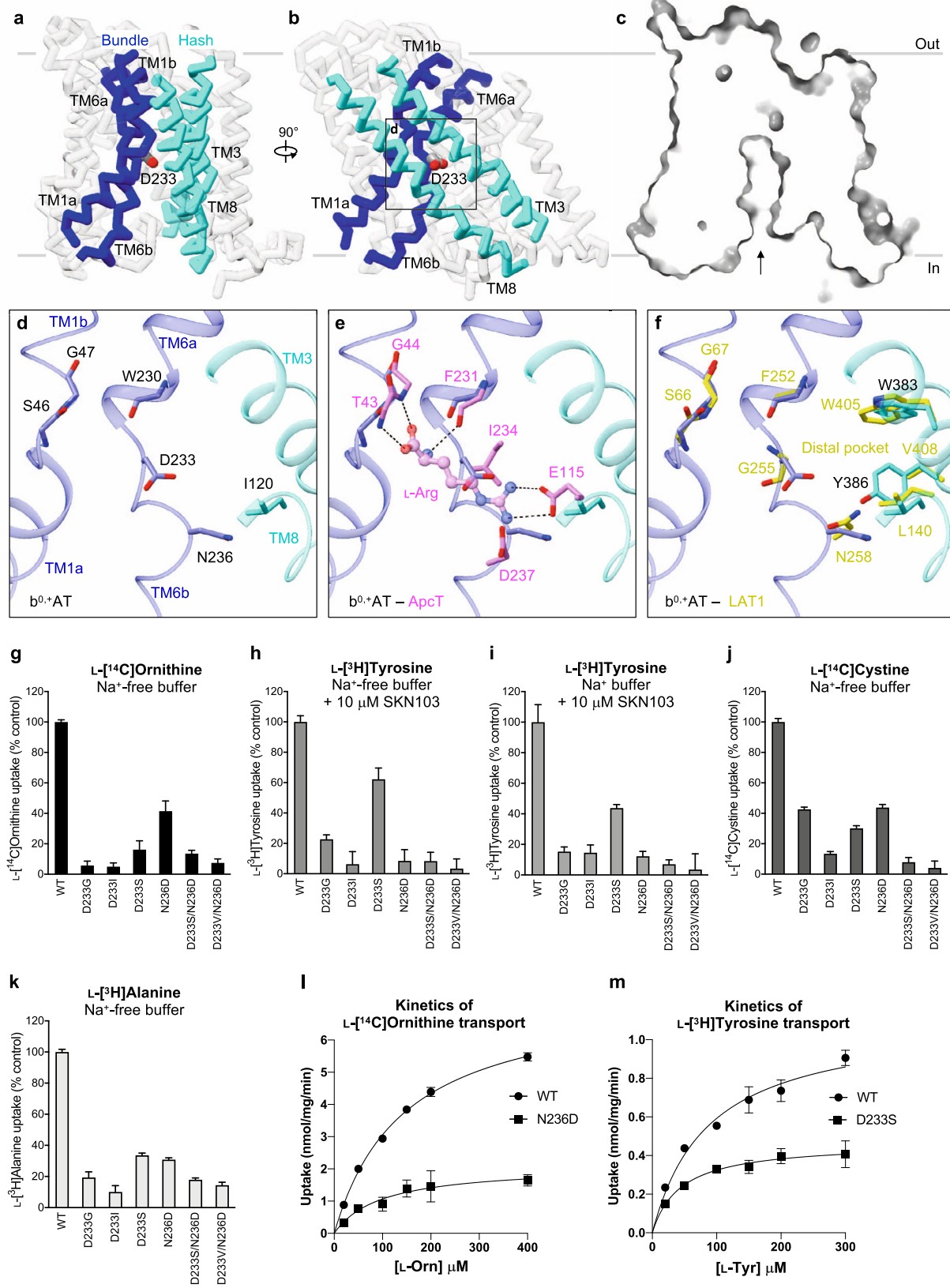

activities for cationic (Orn), neutral (Tyr), small (Ala) amino acids, and cystine (CssC) (Fig. 9g–k). When Asp233 is mutated to neutral amino acids (D233G, D233I, and D233S), b$^{0,+}$AT did not transport Orn (Fig. 9g), confirming that the negative charge of Asp233 interacts with the positive charge of the cationic amino acids, consistent with a recent structure of human b$^{0,+}$AT with Arg

modeled into one of the observed pockets[15]. By contrast, recognition of neutral amino acids did not fully require the negative charge of Asp233, as D233S retained Tyr transport activity (Fig. 9h). The N236D mutation mimics an Asp residue in y$^{+}$L and CATs (Supplementary Fig. 15b), and it retained Orn recognition to some degree but completely lost Tyr transport (Fig. 9g, h),

**Fig. 9 Putative substrate-binding site of b$^{0,+}$AT and transport assays. a, b** Cα trace representation of b$^{0,+}$AT, highlighting the TMs involved in substrate translocation. The hash domain is colored cyan and the bundle domain is blue. The Asp233 is shown as stick model. **c** Cut-away surface representation of b$^{0,+}$AT from the same view as **a**. The inward-facing cavity is indicated by an arrow. **d** Close-up view of putative substrate-binding site. **e** Comparison of b$^{0,+}$AT with ApcT. Residues of ApcT are shown in pink. **f** Comparison of b$^{0,+}$AT with LAT1. Residues of LAT1 are shown in yellow. **g** Uptake of 100 μM L-[$^{14}$C]ornithine in HEK293 cells expressing b$^{0,+}$AT-rBAT or its mutants. Net uptake was normalized to WT set as 100%. **h** Uptake of 100 μM L-[$^3$H] tyrosine by b$^{0,+}$AT mutants in the transfected HEK293 cells. The uptake was performed in Na$^+$-free buffer with 10 μM SKN103 to inhibit LAT1 function. **i** Uptake of 100 μM L-[$^3$H]tyrosine by b$^{0,+}$AT mutants in the transfected HEK293 cells. The uptake was performed in Na$^+$-containing buffer with 10 μM SKN103. **j** Uptake of 50 μM L-[$^{14}$C]cystine by b$^{0,+}$AT mutants in the transfected HEK293 cells. **k** Uptake of 100 μM L-[$^3$H]alanine by b$^{0,+}$AT mutants in the transfected HEK293 cells. **l** Concentration-dependent uptake of L-[$^{14}$C]Ornithine by b$^{0,+}$AT WT and N236D in the transfected HEK293 cells. $K_m$ and $V_{max}$ of L-[$^{14}$C]Orn transport by b$^{0,+}$AT WT are 145 μM and 7.5 nmol/mg prot./min, respectively, those of N236D are 97 μM and 2.1 nmol/mg prot./min, respectively. **m** Concentration-dependent uptake of L-[$^3$H]tyrosine by b$^{0,+}$AT WT and D233S in the transfected HEK293 cells. $K_m$ and $V_{max}$ of L-[$^3$H]Tyr transport by b$^{0,+}$AT WT are 87 μM and 1.1 nmol/mg prot./min, respectively, and those of N233S are 44 μM and 0.5 nmol/mg prot./min, respectively. The values in **g–m** are mean ± SEM. $n$ = 3–4 technical replicates (see data in Source Data file).

supporting its importance for positive charge recognition in y$^+$L and CATs. The profiles of Tyr transport are Na$^+$ independent, confirming the function of system b$^{0,+}$ (Fig. 9h, i)[44]. Kinetics of Orn and Tyr transport by these mutants (N236D and D233S) indicated slightly lower $K_m$ as compared to WT, but largely lower $V_{max}$ values, suggesting similar affinity but altered transport efficiency (Fig. 9l, m). The mutational effects were less prominent in the transport of small amino acids like Ala (Fig. 9k). These results demonstrate the importance of both Asp233 and Asn236 for the selectivity of cationic and large neutral amino acids, and that their functional properties are partially interchangeable among the SLC7 members. Moreover, the interchangeable properties of D233S and N236D imply that system b$^{0,+}$ is more similar to system y$^+$L than to system L or CATs, suggesting an evolutionary significance.

We also measured the transport of cystine, another high-affinity substrate of b$^{0,+}$AT, which causes cystinuria when its transport is impaired. Cystine transport activity was preserved in D233S and N236D, and also highly remained in D233G (Fig. 9j), reminiscent of mixed results for Orn and Tyr transport. The less strict requirement of the tested residues suggests that b$^{0,+}$AT uses multiple residues to recognize cystine. Notably, among SLC7 members, cystine transport is mediated by b$^{0,+}$AT, xCT (SLC7A11)[46], and AGT1 (SLC7A13)[8], which show little conservation in the unwound region of TM6 (Supplementary Fig. 15)[41,45].

Although our single mutants D233S and N236D were designed to mimic system y$^+$L and indeed showed functional properties to partly switch the substrate selectivity (Fig. 9g, h), double mutants (D233S/N236D and D233V/N236D) fully abolished the transport for all substrates (Fig. 9g–k). These results suggest an additional requirement of other residues for the full recognition of the substrate and Na$^+$ in system y$^+$L, such as residues in TM3 and TM8, which are also involved in substrate recognition. Thus, the molecular basis of Na$^+$ dependence in the system of y$^+$L transporters remains an open question.

## Discussion

Although the super-dimerization of system b$^{0,+}$ was reported many years ago[35], its biological relevance has so far remained unclear. Here, using structure-based biochemical assays, we have shown that super-dimerization is a key step for system b$^{0,+}$ biogenesis (Fig. 10a). From the analyses of N-glycan maturation, protein assembly, and localization, we propose that super-dimerization precedes the ER-Golgi trafficking, acting as a checkpoint for protein quality control (Fig. 10a). We also found that super-dimerization is Ca$^{2+}$ dependent, which is plausible given the rich Ca$^{2+}$ pool of the ER lumen, where rBAT folding occurs (Fig. 10a)[12]. Such a role of Ca$^{2+}$ is reminiscent of its known regulatory roles in protein folding machineries and post-translational modifications[33], but the current case is unique in that Ca$^{2+}$ binds directly to the protein being folded. Given that

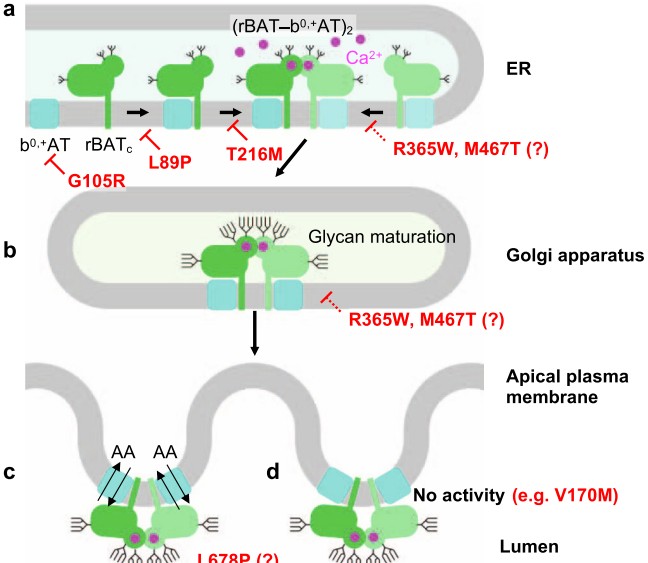

**Fig. 10 A working model for b$^{0,+}$AT-rBAT biogenesis and its defects in cystinuria. a** Model of super-dimer assembly in the ER. The initial step is the heterodimerization of b$^{0,+}$AT and core-glycosylated rBAT (denoted as rBAT$_c$). Since b$^{0,+}$AT is known to be required for oxidative folding and N-glycan maturation of rBAT[12], defects in the b$^{0,+}$AT expression itself (e.g., b$^{0,+}$AT G105R) or the b$^{0,+}$AT–rBAT interaction (e.g., rBAT L89P) would interfere with the initial assembly process and thus prevent glycan maturation. In the latter step, two molecules of b$^{0,+}$AT-rBAT assemble to form a super-dimer, which requires Ca$^{2+}$ binding. Mutations that prevent super-dimerization, such as rBAT T216M and the Ca$^{2+}$-binding site mutants, abolish this step. Incomplete complexes cannot pass the ER quality control. It is noted that rBAT R365W and M467T can form super-dimers to some extent but are probably less stable than the wild-type, due to their positions in the subdomain interfaces. **b** In the Golgi apparatus, the super-dimeric b$^{0,+}$AT-rBAT undergoes full glycan maturation (mature glycans are depicted by longer branches). As rBAT R365W and M467T can form super-dimers, they get partially matured. **c** Upon correct maturation, super-dimeric b$^{0,+}$AT-rBAT will be trafficked to the apical membrane of epithelial cells. Microvilli consist of numerous membrane protrusions, consistent with the curved lipid bilayer of b$^{0,+}$AT-rBAT. **d** Non-type I mutations (e.g., b$^{0,+}$AT V170M) abolish transport activity.

Tyr216′ is in the proximity of the Ca$^{2+}$-binding site (Fig. 4d), a type I cystinuria mutant T216M probably impairs Ca$^{2+}$ binding and affects super-dimerization, preventing its exit from the ER (Fig. 10a). Other cystinuria mutants, in particular R365W and M467T have been known to facilitate protein degradation[12], and our structures suggest that these substitutions affect the domain A–TMD and A–C interfaces, and thus impair its stability

(Fig. 10a, b). Interestingly, b$^{0,+}$AT-G105R showed little expression and resulted in no rBAT maturation (Fig. 10a), suggesting that some of non-type I mutations are, at the molecular level, protein-folding defects. rBAT associates with another SLC7 transporter, AGT1, and the resulting complex has been shown to exist as higher-order assemblies[8]. We therefore speculate that similar Ca$^{2+}$-dependent biogenesis model should apply to rBAT–AGT1.

LeuT-fold transporters are known to operate via a so-called 'rocking-bundle' mechanism, in which the bundle domain undergoes a rocking motion to mediate substrate transport[47]. This major conformational change is associated with minor conformational changes of other structural elements, including EL4a, EL4b, and IL1, which regulate the gating process. In our structure of b$^{0,+}$AT, EL4a and EL4b form a lid above the extracellular gate, interacting with TM1, EL2, and the Aα4-β5 loop (Supplementary Fig. 16a). A similar positioning of EL4a and EL4b is also seen in LAT1[10], albeit with different interactions with CD98hc (Supplementary Fig. 16b). The comparison of outward- and inward-facing structures of bacterial homologs of the SLC7 family, known as AdiC and BasC[48,49] shows that, upon the inward-to-outward structural transition, EL4b dissociates from TM3, and EL4a undergoes upward movement to widen the transport pathway (Supplementary Fig. 16c, d). We speculate that similar movements of EL4a and EL4b would be required for b$^{0,+}$AT to transition to the outward-facing state. Such a movement explains how R365W affects the transport characteristics of b$^{0,+}$AT[50] by altering the Aα4-β5 loop (Fig. 3a; Supplementary Fig. 16a). A similar structural cross-talk could also explain the incomplete negative dominance pattern of some of the non-type I cystinuria[14], where an inactive subunit inhibits the other subunit through the network of homo- and heteromeric interactions.

In conclusion, our study shows that b$^{0,+}$AT–rBAT exists as a dimer of two heterodimers within the membrane and that this assembly is the key to system b$^{0,+}$ biogenesis, and maturation. Although membrane protein oligomerization has been widely observed in the contexts of protein stabilization[51], substrate translocation[47] or lipid ultrastructure[52], its obligatory requirement for N-glycan maturation is so far unique. The strict requirement might reflect the controlled trafficking of system b$^{0,+}$, and AGT1, another rBAT associated transporter, to the apical side of kidney brush-border and intestinal epithelial cells[20]. Our findings also pointed to a previously unknown role of ER Ca$^{2+}$ in the assembly and quality control of system b$^{0,+}$, extending our understanding of its biogenesis model[12,24] and the roles of metal ions in protein assemblies in general. Finally, our study revealed that T216M, one of the most common cystinuria mutations worldwide[1], critically affects the higher-order assembly of the transporter and thereby causes type I cystinuria. Therefore, the restoration of super-dimerization may emerge as a potential therapeutic strategy for curing such types of cystinuria.

## Methods

**Protein expression and purification.** Murine and ovine *SLC3A1* and *SLC7A9* genes encoding rBAT and b$^{0,+}$AT, respectively, were amplified from tissue cDNA (ZYAGEN) and cloned into the pEZT-BM vector (a gift from Ryan Hibbs; Addgene plasmid # 74099)[53]. The cloned ovine *SLC3A1* sequence contained multiple nucleotide polymorphisms as compared to the closest UniProt entry (UniProt ID: W5P8K2). All substitutions are silent except for Arg181′ > Gln181′, which is a non-conserved residue on the rBAT surface and is not relevant for the structures presented in this study. For purification, rBAT was fused with the N-terminal His$_8$-EGFP tag and a TEV cleavage site. b$^{0,+}$AT was fused with the C-terminal TwinStrep II tag. Protein expression was performed as described[53]. Briefly, P2 or P3 baculovirus was produced in Sf9 cells and used to transfect HEK293S GnTI$^-$ suspension cells cultured in Freestyle 293 (Gibco) with 2% FBS at 37 °C with 8% CO$_2$. At the cell density of 2–3 × 10$^6$ cells ml$^{-1}$, the two concentrated viruses were added simultaneously with 5 mM sodium butyrate. After 12–18 hours, the culture temperature was decreased to 30 °C, and cells were harvested 48 hours post-infection.

All purification procedures were performed at 4 °C unless otherwise stated. Harvested cells were resuspended in lysis buffer (50 mM Tris, pH 8.0, 150 mM NaCl and cOmplete protein inhibitor cocktail) and lysed by sonication with a probe sonicator or by homogenization in a glass potter. The membrane fraction was pelleted by ultracentrifugation at 185,000 × g, 1 h 10 min (45Ti, Beckman), resuspended in lysis buffer and stored in aliquots at −80 °C until use. The membrane fraction was solubilized in lysis buffer supplemented with 0.5% lauryl maltose neopentyl glycol (LMNG) and 0.1% cholesterol hemisuccinate (CHS) for 3 h. Insoluble material was removed by ultracentrifugation at 185,000 × g, 30 min (45Ti, Beckman). The supernatant was incubated with GFP nanotrap resin[54] for 2 h, and the resin was washed with ten column volumes of purification buffer (20 mM Tris-HCl, pH 8.0, 150 mM NaCl, 0.01% LMNG and 0.002% CHS). The target protein was cleaved off the column with TEV protease and subjected to size exclusion chromatography on a Superose 6 Increase 3.2/300 column connected to the Ettan system (GE Healthcare) operated at room temperature. The peak fraction was concentrated and used for further analyses (Supplementary Fig. 2a).

For initial negative-stain EM and cryo-EM trials in LMNG, samples were prepared by slightly different protocols. For negative-stain EM, murine and ovine b$^{0,+}$AT constructs were fused with Strep II instead of the TwinStrep II tag and the target complex was purified in two steps, first with the Strep XT resin and then with the GFP nanotrap. For initial cryo-EM in LMNG, b$^{0,+}$AT fused with TwinStrep II was purified in a single-step with Strep XT resin and then subjected to size exclusion chromatography without cutting GFP.

**Liposome assay.** Proteoliposomes were prepared by a published procedure[10]. Briefly, soy PC (40% lecithin) was dissolved in chloroform and dried into a thin film under a continuous flow of nitrogen gas. The film was resuspended into liposome buffer (20 mM HEPES, pH 7.0 and 120 mM NaCl) at a lipid concentration of 20 mg ml$^{-1}$ and sonicated in a bath sonicator for 30 min at room temperature to yield unilamellar vesicles. Purified proteins were added at a protein-to-lipid ratio of 1:166 (w/w) and reconstituted by three freeze-thaw cycles. Reconstituted proteoliposomes were diluted threefold and centrifuged at 8000 × g, 10 min to remove large aggregates, and then the supernatant was ultracentrifuged at 185,000 × g, 2 h to pellet the liposomes. The pellet was stored at −80 °C until subsequent experiments. For uptake experiments, the proteoliposome pellet was resuspended in reaction buffer (20 mM HEPES, pH 7.0 and 120 mM NaCl) with or without 1 mM L-Arg and sonicated with a probe sonicator to form unilamellar vesicles and load the counter-substrates into the liposomes. At this point, small samples were subjected to SDS-PAGE to check the reconstitution efficiency (Supplementary Fig. 2d). Proteoliposomes were passed through a G-50 gel filtration column pre-equilibrated with reaction buffer to remove free L-Arg and then used immediately for transport assays.

To start the transport reactions, proteoliposomes were mixed with four volumes of extraliposomal solution (reaction buffer with 0.11 μM L-[$^3$H]Arg; 54.5 Ci/mmol; PerkinElmer) and incubated at 25 °C. At indicated time points, aliquots were taken and applied to Dowex 50WX4 cation exchange resin in a spin column to remove free L-[$^3$H]Arg. The resin had been pre-equilibrated with reaction buffer and pre-chilled on ice to stop the reaction. After centrifugation at 700 × g, 1 min, the eluted proteoliposomes were mixed with liquid scintillation cocktail (Rotiszint Eco Plus, Carl Roth) and radioactivity was measured with a liquid scintillation counter (Tri-carb 1500, Packard).

**Negative-stain electron microscopy.** After the GFP nanotrap step, purified b$^{0,+}$AT–rBAT samples were diluted to about 0.02 mg ml$^{-1}$ and applied onto glow-discharged copper grids coated with continuous carbon film. Staining was performed with three drops of 1% uranyl formate and blotting with Whatman Grade 50 filter paper. LAT1–CD98hc was prepared as described[10] and stained similarly. Grids were imaged in a Tecnai BioTwin electron microscope operated at 120 kV. Roughly 50 images were recorded for each sample at a 49,000× magnification with a 2.2 Å nominal pixel size and analyzed in RELION 3.0[55].

**Nanodisc reconstitution.** Nanodisc scaffold protein MSP1E3D1 was prepared as described[56]. Briefly, MSP1E3D1 with N-terminal His$_7$ tag and a TEV cleavage site was expressed in *E. coli* BL21 cells. MSP1E3D1 was purified on NiNTA column with four washes with buffer containing Triton X-100 and cholate, and eluted with 400 mM imidazole. The His tag was cleaved by TEV protease, and the sample was passed through a NiNTA column and then dialyzed against storage buffer (20 mM Tris, pH 8.0, and 150 mM NaCl). Purified MSP1E3D1 was concentrated and stored in aliquots at −80 °C.

Nanodisc reconstitution was performed as described[56]. Briefly, the lipid mixture was prepared by dissolving phosphatidylcholine (PC), phosphatidylglycerol (PG), and cholesterol (CHOL) in chloroform at a ratio of 2:2:1 (w/w). The mixture was dried into a thin film in a glass vial under nitrogen gas flow and then resuspended in 50 mM sodium cholate solution to a final phospholipid concentration of 20 mg ml$^{-1}$. Purified b$^{0,+}$AT–rBAT was mixed with MSP1E3D1 and the lipid mixture to a protein:scaffold:lipid ratio of 1:5:200 (mol/mol) adding 20 mM sodium cholate, and incubated at 4 °C with gentle end-to-end rotation. After incubation for 1 h, a first batch of Bio-Beads SM-2 resin (500 mg per 1 ml mixture) was added to start detergent removal. After 2 h, a second batch was added, and the detergent

removal proceeded overnight. The reconstituted nanodiscs were recovered by passing through a spin column and subjected to size exclusion chromatography on a Superose 6 Increase 3.2/300 column at room temperature. Peak fractions were concentrated to about 10 mg ml$^{-1}$ and immediately used for cryo-EM grid preparation.

**Sample vitrification for cryo-EM.** All cryo-EM grids used in this study were Quantifoil 400 copper R1.2/1.3 grids. Grids were pre-washed in acetone and dried for at least 1 h and then glow-discharged twice in a Pleco glow discharger under 0.4 mBar air at 15 mA for 45 seconds. For vitrification, a Vitrobot Mark I was set to 6 °C and 100% humidity, and filter paper for blotting (Whatman Grade 595) was pre-equilibrated in the chamber for about 1 h. 3 μl of protein solution at 10 mg ml$^{-1}$ were applied to the grids, blotted for 3–4 s with blot force 20, and plunge-frozen. To improve sample distribution, 1.5 mM fluorinated Fos-Choline-8 was added to the sample immediately before vitrification. For detergent and nanodisc dataset #1, we added 10 mM substrate Arg about 30 min before vitrification, hoping to visualize the bound substrate, but none of the resulting maps showed any substrate density, indicating that the protein was in the apo state.

**Cryo-EM data collection.** All cryo-EM data were recorded on Titan Krios microscopes operated at 300 kV. For b$^{0,+}$AT–rBAT in LMNG, the data were recorded on Titan Krios G3i at 105,000 × magnification with a K3 direct electron detector operated in a counting mode, resulting in a calibrated pixel size of 0.837 Å. The electron exposure rate was set to 15 e pix$^{-1}$ s$^{-1}$, and the movies were recorded for 2 s with 50 frames, resulting in a total exposure of 50 e Å$^{-2}$ on the specimen. For b$^{0,+}$AT–rBAT in nanodisc, data were recorded in three separate sessions. In the first session, settings were as for the LMNG dataset. In the second session, similar settings were used on another Titan Krios G2 microscope, equipped with a K3 camera with a calibrated pixel size of 0.831 Å. In the third session, a Titan Krios G2 with Falcon III camera was used. For the Falcon III data collection, the magnification was set to 96,000 ×, resulting in a calibrated pixel size of 0.833 Å, and the exposure rate was set to 1.5 e pix$^{-1}$ s$^{-1}$. Each movie was recorded for 30 s with 50 frames in a counting mode, resulting in a total exposure of 40 e Å$^{-2}$. All movies were saved in gain-normalized TIFF format as specified by the EPU software (Thermo Fischer Scientific).

**Image processing.** Single-particle image processing was performed in RELION 3.0[55] (Supplementary Fig. 3a,b). For the nanodisc datasets, data from three sessions were processed separately to the particle polishing step. Beam-induced motion was estimated and corrected in MotionCor2[57] using 5 × 5 patches with dose-weighting. Defocus and astigmatism were estimated in Gctf4[58]. Initial particle sets were picked by using a Laplacian-of-Gaussian filter from a few dozen micrographs and extracted with a box size of 330 pix down-sampled to 80 pix. After 2D and 3D classifications, the resulting 3D map served as a template for automated picking in RELION. In parallel, coordinates from the resulting 3D classes were used to train Topaz[59] for neural network-based picking. The Topaz network was trained on dataset #1 with 8 × down-sampled micrographs and around 6000 input coordinates. Particles picked by the two programs were extracted separately with down-sampling from 330 pix to 80 pix and subjected to rounds of independent 3D classification. After removing duplicate particles, good particles were re-extracted with a box size of 420 pix down-sampled to 320 pix, resulting in 1.099 Å/pix for nanodisc dataset #1. Particles were refined with an auto-refine procedure with a soft mask and C2 symmetry imposed. After the initial 3D refinement, particle trajectories and weighting B/k-factors were refined by Bayesian polishing[60], and then per-particle defocus values were refined as implemented in RELION 3.0[55]. For the detergent dataset, CTF-refined particles yielded the final reconstruction at 3.91 Å resolution according to the FSC = 0.143 criterion (Supplementary Fig. 3a,d).

For the nanodiscs datasets, CTF-refined particles were further processed in RELION 3.1, to accommodate the different pixel sizes in different datasets[61]. Particles from the three datasets were merged with different rlnOpticsGroup labels, and then refined against the reference map of dataset #1 (Supplementary Fig. 3c). Resulting alignments and volumes were used to correct possible errors in pixel sizes by using anisotropic magnification correction, which reported less than 0.5% errors in the assumed pixel sizes. The 3D auto-refinement from the merged particles after magnification correction with C2 symmetry yielded a 2.88 Å map of the full complex in lipid nanodisc (Supplementary Fig. 3c–e).

The map of the full complex indicated structural flexibility between the two b$^{0,+}$AT–rBAT protomers. To improve local map quality, we took two approaches: (i) focused refinement for the rBAT ectodomain and (ii) multi-body refinement of individual b$^{0,+}$AT–rBAT subcomplexes. For focused refinement, the TMD signal was subtracted from individual particles, and then subtracted particles were refined against the rBAT ectodomain map with C2 symmetry imposed, yielding a 2.68 Å resolution map for the rBAT ectodomain (Supplementary Fig. 3e). Multi-body refinement was performed with two bodies, each covering an individual b$^{0,+}$AT–rBAT protomer and half the micelle. Bayesian priors were set to 10 degrees for body orientations and 3 pixels for translations. After multi-body refinement, we merged the two bodies to obtain a single reconstruction for the b$^{0,+}$AT–rBAT heterodimer subcomplex, since both bodies consist of identical sets

of subunits. To do this, signal subtraction was performed for each body as described[62], and the resulting particles were merged and subjected to rounds of 3D classification to further select good particles, and refined against a map of the b$^{0,+}$AT–rBAT heterodimer. Final refinement yielded a 3.05 Å resolution map for the b$^{0,+}$AT–rBAT heterodimer subcomplex with improved local resolution for the TMD (Supplementary Fig. 3f).

To visualize the major conformational heterogeneity in the dataset, we performed a principal component (PC) analysis. We then divided the particles into three subsets of roughly equal size along the PC1 axis and performed normal 3D refinement for each subset. Morphing of the resulting three maps revealed a flexible motion of two b$^{0,+}$AT molecules relative to the more rigid rBAT–rBAT assembly (Supplementary Movie 1).

**Model building and refinement.** The initial model for b$^{0,+}$AT–rBAT was generated by homology modeling using MODELLER 9.22[63]. Homologous proteins were first identified using BLAST, and top hits were selected to cover the entire sequence of rBAT. Three structures were thus selected and used for homology modeling, namely *Bacillus cereus* oligo-1,6-glucosidase (PDB 1UOK), *Xanthomonas campestris* alpha-glucosyl transfer enzyme (PDB 6AAV), and LAT1–CD98hc (PDB 6IRS). The input multiple sequence alignment generated in Clustal Omega[64] covered almost all regions except for the C-terminal peptide of rBAT (residues 655–685). The MODELLER parameters were set as follows: library_schedule = autosched.slow; max_var_iterations = 300; md_level = refine.slow; repeat_optimization = 2; starting_model = 1; and ending_model = 5. All five resulting models MODELLER showed good scores with slightly different orientations between ED and TMD. One of them was used for further model building.

All manual model building was performed in COOT 0.8.9[65]. The cryo-EM maps resolved all residues except for the disordered terminal residues (residues 1–28 and 484–487 for b$^{0,+}$AT, and 1–62 for rBAT). After building the protein, prominent densities on the predicted N-glycosylation sites were built as acetyl glucosamine units (GlcNAc). A spherical density within domain B was built as a Ca$^{2+}$ ion (see Main Text). Among the numerous lipid densities resolved around the TMD, two flat-shaped densities in the outer leaflet were assigned as cholesterols. One density near TM1′, which had two elongated tails, was modeled as a phosphatidylcholine. Both lipids had been included in the lipid nanodisc composition (see Nanodisc reconstitution in Methods). Ligand restraints were generated in AceDRG[66] and applied throughout manual model building and refinement.

All model refinements were performed in phenix.real_space_refine in PHENIX (version 1.16-dev3689)[67]. After iterative refinement of b$^{0,+}$AT–rBAT against the heterodimer map, the model of the rBAT ectodomain was independently refined against the rBAT ectodomain map with a refinement resolution set to 2.6 Å. The refined rBAT ectodomain model was then merged back to the heterodimer model and refined again. For the full complex map, the two heterodimer models were fitted as rigid bodies. Model validation was performed with MolProbity[68]. Data collection and validation statistics are shown in Supplementary Tables 1 and 2.

For structure visualization, UCSF ChimeraX and the associated software packages were used[69]. The electrostatic potentials were calculated with APBS[70]. Multiple sequence alignments were generated with Clustal Omega[64], and formatted with ESPript 3[71].

**Endo H assay.** The Endo H assay was performed as described previously[12] with small modifications. One modification was that we used fluorescent tags instead of using Western blotting. This not only simplified the experimental procedures, but also allowed dual-color fluorescence detection to visualize the heteromeric complexes on SDS-PAGE gels directly. For the assay, the pEZT-BM expression plasmids as described above were adopted with minimal modification. The rBAT plasmid was identical to that used for structural analysis (with the N-terminal GFP tag; pEZT-NGFP-OaSLC3A1 plasmid) and the b$^{0,+}$AT was fused with an additional mCherry tag on the C-terminus after TwinStrep II (the pEZT-C-TwinStrep-mCherry-OaSLC7A9 plasmid).

Protein co-expression was performed in HeLa cells cultured at 37 °C, 8% CO$_2$ in DMEM supplemented with 10% FBS (Gibco) and penicillin/streptomycin mixture (Pen Strep, Gibco). Two days before transfection, cells were seeded onto 24-well plates at 0.3 × 10$^6$ cells/well. Prior to the transfection, 500 μl of the fresh medium were exchanged. Plasmids coding b$^{0,+}$AT and rBAT (250 ng each) were mixed into 100 μl of DMEM, and 1.5 μl of Polyjet reagent in 100 μl DMEM were added. After incubation at room temperature for 15–20 min, DNA:polyjet mixture was added dropwise to the HeLa cell culture. Fresh medium was exchanged at 12–18 h post transfection. The cells were harvested at 48 h post transfection, and cell pellets were frozen.

Pelleted cells were thawed and washed once with lysis buffer (50 mM Tris-HCl, pH 8.0, 150 mM NaCl and cOmplete protease inhibitor), re-pelleted, and re-suspended in 100 μl lysis buffer. Cells were lysed by brief sonication with a probe sonicator (0.5 s pulse and 2 s interval, 10 cycles) and the cell lysate was directly used for the assay. The glycosidase reaction was performed with Endo H enzyme (NEB) under non-denaturing conditions following manufacturer's instructions. Briefly, 5 μl of cell lysate was added with 0.2 μl Endo H enzyme in Glyco 3 buffer. The

reaction was performed incubated for 5 h at 37 °C. Negative controls were reactions without Endo H enzyme. The reactant was mixed with the 2 × SDS-PAGE sample buffer containing 100 mM β-mercaptoethanol and run on 12% Mini-PROTEAN TGX Precast Gel (Bio-Rad) without a boiling step. The gels were directly imaged on Gel Analyzer (Bio-Rad). Fluorescent wavelengths and exposure times were set as follows: GFP (488 nm, 120 s), mCherry (546 nm, 1440 s) and pre-stained marker (680 nm, 2 s).

**Site-specific Cys cross-linking assay.** Protein co-expression and cell lysis were performed as in the Endo H assay. To prevent spontaneous reduction of disulfide bonds after cell lysis, lysed cells were first treated with 0.5 mM 4-DPS at 4 °C for 2 min to generate oxidizing conditions. Then, to prevent non-specific disulfide bond formation after SDS denaturation, free cysteines were blocked by treating with 500 mM iodoacetamide at 4 °C for 5 min. The treated lysates were subjected to 12% Mini-PROTEAN TGX Precast Gel (Bio-Rad) with or without reducing agent. Gels were imaged in the same way as in the Endo H assay.

**Fluorescence-detection size exclusion chromatography of mutants and with EGTA.** Protein stability was analyzed by fluorescence-detection size exclusion chromatography (FSEC) using the N-terminally GFP-tagged rBAT and the C-terminally mCherry-tagged b$^{0,+}$AT. Proteins were expressed transiently in HEK293S GnTI$^-$ cells with the same protocol as for the Endo H assay. Cells were lysed by sonication and solubilized in 50 mM Tris-HCl (pH 8.0), 150 mM NaCl, 1% LMNG, and 0.1% CHS at 4 °C for 2 h. The insoluble materials were removed by ultracentrifugation and the supernatant was subjected to FSEC on a Bio SEC-5 column (4.6 × 300 mm, pore size 300 Å, Agilent) equilibrated with 10 mM HEPES, 150 mM NaCl and 0.03% n-Dodecyl-β-D-maltoside. Fluorescent signals were recorded on a Shimadzu SPD-20AV UV-VIS detector for GFP (512 nm) and mCherry (610 nm). For analyzing the effect of a Ca$^{2+}$ chelator, 10 mM EGTA was added after protein solubilization, and the samples were heated for 10 min on a block incubator at indicated temperatures before being subjected to FSEC. Melting curves were fitted using Fityk software[72].

**Localization of rBAT and b$^{0,+}$AT.** Localization of rBAT and b$^{0,+}$AT was examined in HeLa cells. The cells (1.0 × 10$^5$ cells/well) were seeded on collagen-coated coverslips in a 6-well plate 24-hour before transfection. Wild-type or mutant clones encoding rBAT (pEZT-NGFP-OaSLC3A1 plasmid) and b$^{0,+}$AT (pEZT-C-TwinStrep-mCherry-OaSLC7A9 plasmid) at 1:1 molar ratio were transfected into the cells by using PEI MAX (40,000 MW Linear; Polysciences, Inc.) using DNA: PEI ratio of 4 μg total DNA: 8 μg PEI. Transfected cells were continuously cultured for 2 days. For imaging, the cells were washed with PBS containing 0.1% Tween-20 (PBS-T), fixed with 4% w/v paraformaldehyde in PBS for 15 min at room temperature, and washed with PBS again. Nonspecific signals were blocked by Blocking One Histo (Nacalai Tesque) for 30 min at room temperature. The cells were stained with either ER marker anti-PDI (C81H6 lot. 3; Cell Signaling#3501 S) or Golgi apparatus marker anti-RCAS1 (D2B6N lot. 4; Cell Signaling#11290 S) antibodies diluted in Can Get Signal B (Toyobo, Japan) with ratio 1:240 for overnight at 4 °C. After washing with PBS, samples were incubated with the secondary antibodies conjugated with Alexa Fluor 647 and DAPI (1 ng; for nucleus staining) diluted (1:240) in Can Get Signal B for 1 h at room temperature. After PBS wash, specimens were mounted with ProLong Glass Antifade Mountant (Invitrogen). The images were taken with confocal microscopy (Zeiss LSM880, Alpha Plan Apochromat 1.46 NA, 100× objectives). Data were processed using Z-stacks with the optimal z-sectioning thickness around 300 nm, followed by post-processing using the provided algorithm from the ZEISS LSM880 platform.

**Transport assay in transfected cell lines.** Genes encoding rBAT and b$^{0,+}$AT were cloned into the pEG BacMam vector (a gift from Eric Gouaux[73]) without a tag. Transport functions of rBAT and b$^{0,+}$AT mutants were measured in HEK293 cells. The cells were seeded on poly-D-lysine-coated 24-well plate at 1.7 × 10$^5$ cells/well and cultured in DMEM containing 10% FBS, at 37 °C and 5% CO$_2$. Transfection was performed at 24 h after seeding. Wild-type or mutant clones encoding rBAT (pEG-OaSLC3A1 plasmid) and b$^{0,+}$AT (pEG-OaSLC7A9 plasmid) were transfected at 1:1 molar ratio into HEK293 cells by using lipofectamine 3000 (Thermo) following the manufacturer's protocol. Two days transfection, transport assays were performed as described[74]. Briefly, transport of 100 μM L-[$^{14}$C]ornithine (2 Ci/mol; Moravek Biochemicals), 50 μM L-[$^{14}$C]cystine (1 Ci/mol; PerkinElmer), 100 μM L-[$^3$H]tyrosine (10 Ci/mol; PerkinElmer) and 100 μM L-[$^3$H]alanine (5 Ci/mol; Moravek Biochemicals) were measured for 3 min in Na$^+$-free HBSS pH 7.4. Transport of L-[$^3$H]tyrosine was assayed in the presence of 10 μM SKN103 to inhibit endogenous LAT1 function[75]. For uptake of L-[$^3$H]tyrosine in Na$^+$-containing condition, HBSS buffer was used. After terminating the reaction and cell lysis, an aliquot of the lysate was used to measure protein concentration by BCA protein assay (Takara Bio). The lysate was mixed with OptPphase HiSafe 3 (PerkinElmer), and radioisotope activity was monitored by LSC-8000 β-scintillation (Hitachi). Data shown in the figures were those subtracted by the

uptake values in Mock cells (the cells transfected with empty plasmids). Kinetics of L-[$^{14}$C]ornithine and L-[$^3$H]tyrosine transport were determined for 3 min using radioisotope substrates at a concentration of 20–400 μM. Curves were fitted to Michaelis–Menten plots (GraphPad Prism 8.4).

**Pulse-chase experiment.** Pulse-chase experiments were performed following Bartoccioni et al. with modifications[12]. HeLa cells on 35-mm plates were transfected with pEG-OaSLC7A9-WT and pEG-NGFP-OaSLC3A1-WT or its Ca$^{2+}$-binding site mutants for 18 h. The cells were washed with PBS and incubated in methionine/cystine-free MEM containing dialyzed FBS for 30 min followed by the addition of EasyTag Express [$^{35}$S] Protein Labeling mix (100 μCi/ml; PerkinElmer) for 30 min. After pulse, the media was removed and the cells were incubated with MEM containing FBS, 5 mM methionine and 5 mM cysteine in the presence or absence of 100 nM thapsigargin (Tg). At the indicated chase time, the cells were washed twice with PBS and harvested.

GFP-nanobody conjugated with sepharose resin was used for immunoprecipitation. N8his-GFPenhancer-GGGGS4-LaG16 plasmid expressing GFP-nanobody was a gift from Motoyuki Hattori[76]. GFP-nanobody was expressed and purified by using TALON Superflow (Clontech) following the previous publication[76]. Purified GFP-nanobody were conjugated with NHS-activated sepharose 4 Fast Flow following the manufacture's protocol (Cytiva).

Cell pellets were lysed for 2 h in the buffer containing 20 mM Tris-HCl pH 7.4, 150 mM NaCl, 10% v/v glycerol, 20 mM N-ethylmaleimide (NEM), cOmplete protease inhibitor and 2% w/v Fos-choline-12 (FC-12; Anatrace). The supernatant was collected after 20,000 × g centrifugation. Immunoprecipitation was performed by adding the pre-washed GFP-nanobody conjugated sepharose resin to the total lysate. After binding for 2 h at 4 °C, the resin was washed 5 times with the above buffer but contains only 0.1% FC-12. The proteins were eluted by non-reducing SDS-PAGE sample buffer and all eluent was subjected to 7% non-reducing SDS-PAGE. The gels were vacuum-dried at 75 °C for 6 h and then exposed on an imaging plate BAS 2040 (Fujifilm) for 3 days. Radiation image was detected using the phosphorimager Typhoon FLA 7000.

**Cycloheximide-chase experiment.** HeLa cells on 35-mm plate were transfected with pEZT-C-TwinStrep-mCherry-OaSLC7A9 and pEZT-NGFP-OaSLC3A1 plasmids or its mutants for 12 h. The cells were incubated in the media containing 50 or 100 μg/ml cycloheximide. The cells were washed twice and collected at the indicated chase time. Total lysate was prepared as described in the pulse-chase experiment. Protein concentration was measured by BCA assay and the equal amounts of the samples were applied on 7% non-reducing SDS-PAGE. GFP and mCherry fluorescence signals were detected on ChemiDoc MP Imaging System (Bio-Rad).

**Thapsigargin treatment.** HeLa cells on 35-mm plate were transfected with pEZT-C-TwinStrep-mCherry-OaSLC7A9 and pEZT-NGFP-OaSLC3A1 plasmids or its mutants for 12 h. For the negative control, pEG-mCherry-SLC7A5 plasmid (encoding hLAT1) and pEG-NGFP-SLC3A2 plasmid (encoding hCD98hc) were used[10]. The cells were incubated in the media containing thapsigargin at the indicated concentrations for 6 h. The cells were washed twice and harvested. Total lysate was prepared as described in the pulse-chase experiment and the equal amount of protein samples were applied on 7% non-reducing SDS-PAGE. GFP and mCherry fluorescence signals were detected on ChemiDoc MP Imaging System (Bio-Rad).

**Sampling, statistics and reproducibility.** The cell lines used in all biochemical experiments displayed appropriate morphologies and growth during all experimental procedures. Unless otherwise indicated, all experiments were performed at least 3 times independently from different cell passages and showed similar results. Displayed Figures were derived from representative data of reproducible experiments. Band intensities from the gel images were analyzed by Image Lab 6.0.1 (Bio-Rad). Transport assays in the transfected cells were performed in 3–4 technical replicates. The experiments were repeated 2–3 times independently using different cell passages and showed similar results. All graphs and statistical analyses were plotted by GraphPad Prism 8.4.

**Reporting summary.** Further information on research design is available in the Nature Research Reporting Summary linked to this article.

## Data availability

The atomic coordinates have been deposited to Protein Data Bank under accession numbers 7NF6 (b$^{0,+}$AT–rBAT heterodimer), 7NF7 (rBAT ectodomain) and 7NF8 (full complex composite model). Cryo-EM maps have been deposited to Electron Microscopy Data Bank under accession numbers EMD-12296 (b$^{0,+}$AT–rBAT heterodimer), EMD-12297 (rBAT ectodomain), and EMD-12298 (full complex). Unprocessed gel pictures and raw data for all graphs were included in the Source Data file, provided with this paper. All other data will be available upon request.

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

## Acknowledgements
We thank Susann Kaltwasser, Simone Prinz, Mark Linder and Sonja Welsch in the Central Electron Microscopy Facility of Max Planck Institute of Biophysics for technical assistance in electron microscopy; Sabine Häder, Christina Kunz and Heidi Betz for help in lab experiments; Juan Castillo, Özkan Yildiz, the Central IT team and the Max Planck Computing and Data Facility for maintaining the computational infrastructure; David Wöhlert and Martin Centola for technical advice and help in mammalian cell culture and protein expression; Department of Molecular Membrane Biology, Max Planck Institute of Biophysics for sharing HeLa cell culture; Yoko Tanaka and Reiko Kimura for technical assistance; Haruka Minowa in the Radioisotope Research Facilities of The Jikei University for the facilities in pulse-chase experiments; Genrou Kashino for Radioisotope Research facility at Nara Medical University; Motoyuki Hattori at Fudan University for GFP-nanobody plasmid; Pattarawut Sopha at Chulabhorn Graduate Institute for technical advice; and Tomokazu Matsuura in Department of Laboratory Medicine at The Jikei University for his great support. This work was supported by the Max Planck Society to Y.L., D.J.M. and W.K., the Kazato Research Foundation (Kazato Research Encourage Prize 2021), JSPS KAKENHI (JP21K15031) and JST, PRESTO (Grant Number JP21459588) to Y.L., JSPS KAKENHI (JP21H03365) and AMED (Grant Numbers JP21gm0810010, JP21ek0310012, and JP21lk0201112) to S.N., and JSPS KAKENHI (JP19K07373) and Mochida Memorial Foundation grant for Medical and Pharmaceutical research to P.W. Y.L. was supported by Toyobo Biotechnology Foundation Fellowship and Human Frontier Science Program Long-Term Fellowship.

## Author contributions
Y.L. and S.N. initiated the study. Y.L. performed structural studies and in vitro biochemical experiments. P.W. performed in vitro biochemical experiments. P.W., P.K., and S.M. performed fluorescence imaging. D.M. maintained and aligned the electron microscopes and directed data collection. S.N. and W.K. directed and supervised the project. Y.L., P.W., and S.N. wrote the manuscript, with contributions from W.K.

## Funding

## Competing interests
The authors declare no competing interests.
