## [Peer Review File · Nature Communications]

REVIEWER COMMENTS

Reviewer #1 (Remarks to the Author):

The study by Lee et al reports the cryo-EM structure of the cystine/dibasic amino acid exchanger (rBAT/b⁰,+AT) and reports on experiments aimed at understanding the folding and trafficking of this protein in the cell. The structural work is of a high standard and I have no concerns regarding the quality and accuracy of the reported structure. My only comment here is whether the authors can categorically claim the bending in the nanodisc is real or artefactual. The authors support the bending observation by saying that rBAT/b⁰,+AT is found in the microvilli of intestinal cells. However, rBAT/b⁰,+AT is also found in non-polarised cells, where presumably the curvature is considerably less. I just wonder whether the authors are overinterpreting this aspect of the structure.

The main novelty reported in the current study is the role of calcium in coordinating the dimer interface between the two rBAT subunits of the dimer and the functional assays that explore substrate specificity.

Overall I think the study is interesting and tackles an often overlooked part of transport regulation in the cell, i.e. trafficking and localisation. However, the data for the trafficking aspect of the study in my opinion lacks the in depth analysis required to explain the role of the dimerisation in protein maturation.

Firstly, my understanding is that the observation that rBAT needs to dimerise to correctly traffick the super-complex is novel. Previous work in this area had only identified that the rBAT/b⁰,+AT dimer needed to form to initiate trafficking. Secondly, this study may explain the previous observations that rBAT needs to associate with b⁰,+AT to fold correctly (see Rius et al, JBC 287 p22 2012). However, the current assay, as I detail below feels a little superficial and unable to really address what is happening in the cell at a mechanistic level.

I would have liked to see localisation data for the different mutations reported in Figure 5d and 5h. Essentially the main novelty in this report are the insights gained into the role of rBAT in trafficking the dimer of heterodimers ([rBAT/b⁰,+AT]₂) in the cell. Although I applaud the authors in their structural work, I didn't see anything novel in this study from a structural perspective beyond the previous Sci Adv paper by Yan et al (DOI: 10.1126/sciadv.aay6379), except the identification for a role of calcium in dimer formation, which pertains to the trafficking.

The EndoH assay is rather low resolution from a mechanistic perspective - essentially it distinguishes those proteins that have trafficked to the PM from those that haven't. It doesn't tell us anything about what is happening in the cell. For example, do some of these mutants get trapped in different parts of

the ER or Golgi? Are they degraded at different rates? I don't think this is an unfair comment, as the authors refer to degradation in Figure 7. But this rather a catch all term - what route of degradation, on what time scale. Answers to these questions would address a mechanistic view of the phenotypes seen in patients heterozygous for trafficking mutations.

I felt the trafficking analysis needed more analysis and work. The current results definitely prove that rBAT dimerisation is required for the efficient transport of ([rBAT/b_{0,+}AT]₂) to the PM, however, that was already known (see Fernandez et al. JBC 281, p36 (2006). Unless the authors can argue that the role of calcium here is regulatory?

At the very least I would also like the authors to run FSEC analyses on the mutants used to demonstrate the proteins are still stable and that the negative glycosylation result is not due to protein misfolding/aggregation as opposed to aberrant trafficking.

The insights gained from the transport assays is also novel and well executed. The authors main point here is that Asp233 and Asn236 are important for the selectivity of amino acids in this system. This observation builds on previous work in the literature. In addition, the authors suggest that system b_{0,+} is more similar to y⁺L than system L or the CATs, which is functionally interesting. The assay data is good, however, I would like to see SDS-PAGE gels for each of the mutants to show that reconstitution was the same for each sample analysed in Figure 6g.

Reviewer #2 (Remarks to the Author):

The manuscript submitted by Lee and colleagues uncovers one of the molecular mechanism leading to Cystinuria as a consequence of congenital mutations localized in rBAT, the beta subunit of the amino acid transporting system b_{0,+}, composed by the heterodimerization of rBAT and the amino acid transporter b_{0,+} AT. As in the previous structures of rBAT-b_{0,+} AT, here the authors identified a Ca²⁺ ion coordinated by residues located at the interface that bridges two r-BAT/b_{0,+} AT heterodimers, thus stabilizing a supercomplex formed by the two heterodimers. Previous studies (including this one) demonstrated that the formation of this supercomplex is pivotal for protein maturation and trafficking. In this work, the authors demonstrate using functional and biochemical experiments that Cystinuria mutations that perturbs this Ca²⁺ binding site at the dimer-dimer interface also impair both, the formation of this supercomplex and protein maturation resulting in a defect of trafficking of the complex to the final membrane destination.

This work provides a step ahead on our understanding of the molecular mechanisms leading to Cystinuria. Combining structural and biochemical data, the authors establish solid links between rBAT-b_{0,+} AT maturation and r-BAT/b_{0,+} AT supercomplex formation where the Ca²⁺-mediated r-BAT-rBAT dimerization plays a leading role, thus uncovering the molecular mechanism leading to Cystinuria of rBAT variants found in Cystinuria patients. This information is also relevant for future development of therapeutic strategies to cure Cystinuria. Moreover, the conclusions expressed by the authors are well supported by the experimental data. Therefore, in my opinion, this manuscript reaches the standards for publication in Nature Communication and therefore I recommend it for publication. Nevertheless, some questions and remarks need to be addressed by the authors before the final acceptance of this manuscript.

1. It is important that the authors discuss further about the overall conformation of the ovine r-BAT/b_{0,+} AT structure obtained in their work and compare it with the previous two structures of human r-BAT/b_{0,+} AT. For instance, is b_{0,+} AT in the same conformation as the previous structures ?
2. To better understand the objective of the EndoH experiments, the authors need to explain better the assay. For instance, fully mature glycosylated protein is resistant to EndoH treatment.
3. Is the formation of the supercomplex affected at some extent by the presence of the fused fluorescent proteins ?
4. In line 269 the authors state that a decrease in maturation due to mutations at positions N214, D284 or E321 leads to a decrease on Ca²⁺ binding. While theoretically these mutations could disturb the Ca²⁺ binding site, the authors do not provide experimental evidences of this (e.g, a Ca²⁺ binding assay ?). For instance, have the authors tested the effect of Ca²⁺ chelating agents on the supercomplex formation after protein purification? . Please, can you comment ?
5. Line 289. What are the experimental evidences supporting that G105R mutation in b_{0,+} AT leads to improper protein folding ?.
6. Lines 310-312 and Figure 5h. In lanes 3-5 in Figure 5h, the intensities of the bands of these mutants are much weaker than WT, even at the gel bands assigned to each monomeric protein (also see in Supplemental Figure S10). Therefore, other mechanisms other than the disruption of the superdimer could be envisaged to explain the impairment of protein expression caused by these mutations ?. If this is the case, the authors should comment this in the main text.
7. Line 336 and 354. These sentences should be reconsidered as this does not seem true for the L89P mutant.

Minor points

Line 64 . Please, give more details concerning the sentence "...the insertion of 110 residues...". In particular, in what part of the protein is found this extra region ?.

Line 86 . The term "trafficking disease" could be misleading. Can the authors confirm that this term is commonly used in the field ? .

Line 92. Please change “ good solution behavior in detergent” by “good behavior in detergent solution”

Line 374. In the absence of an experimental evidence I suggest to change “...indicating that...” by “suggesting that..”.

Fig 7a. There is a missing reference.

Line 459. I suggest to describe AdiC and add a reference.

Response to referees

Reviewer #1 (Remarks to the Author):

The study by Lee et al reports the cryo-EM structure of the cystine/dibasic amino acid exchanger (rBAT/b⁰,+AT) and reports on experiments aimed at understanding the folding and trafficking of this protein in the cell. The structural work is of a high standard and I have no concerns regarding the quality and accuracy of the reported structure. My only comment here is whether the authors can categorically claim the bending in the nanodisc is real or artefactual. The authors support the bending observation by saying that rBAT/b⁰,+AT is found in the microvilli of intestinal cells. However, rBAT/b⁰,+AT is also found in non-polarised cells, where presumably the curvature is considerably less. I just wonder whether the authors are overinterpreting this aspect of the structure.

Answer:

We thank the reviewer for overall positive comments on the structural aspect of our work. As for the critical comment on the interpretation of membrane bending, we fully agree that the current evidence is too weak to claim any solid connection between the bending observed here and its functional importance in biological context. To draw attention and remain neutral about this controversial yet interesting point, we added a new sentence in the main text.

- Lines 138-141, Page 6:

“However, given that b⁰,+AT-rBAT is also found in non-polarized cells where membrane curvature is presumably less, the lipid bending observed here could be specific to our sample preparation condition, and its physiological relevance needs further clarification.”

The main novelty reported in the current study is the role of calcium in coordinating the dimer interface between the two rBAT subunits of the dimer and the functional assays that explore substrate specificity.

Overall I think the study is interesting and tackles an often overlooked part of transport regulation in the cell, i.e. trafficking and localisation. However, the data for the trafficking aspect of the study in my opinion lacks the in depth analysis required to explain the role of the dimerisation in protein maturation.

Answer:

We appreciate that the reviewer found our study interesting and provided critical comments on the trafficking aspect. We too think this is the most important part of our study. Therefore, we have taken these comments seriously and performed extensive experiments on protein localization and dimerization, as detailed in the answers below.

Firstly, my understanding is that the observation that rBAT needs to dimerise to correctly traffick the super-complex is novel. Previous work in this area had only identified that the rBAT/b^{0,+}AT dimer needed to form to initiate trafficking. Secondly, this study may explain the previous observations that rBAT needs to associate with b^{0,+}AT to fold correctly (see Rius et al, JBC 287 p22 2012). However, the current assay, as I detail below feels a little superficial and unable to really address what is happening in the cell at a mechanistic level.

Answer:

We admit that our original manuscript could not address at mechanistic detail what is happening in the cell during protein assembly and trafficking. To address these in more detail, we performed two new experiments.

First, we performed pulse-chase assays using radio-labeled amino acids and monitored the behaviors of different assembly intermediates in a physiologically realistic time scale (~6 hours). Our pulse-chase experiments using rBAT-WT or its Ca²⁺-binding site mutants clarify that core-glycosylated rBAT (rBAT_c) synthesis and b^{0,+}AT-rBAT_c hetero-dimerization occur first, and super-dimerization happens next. Interestingly, addition of an ER calcium pump inhibitor, Thapsigargin, led to the loss of super-dimers, while retaining the hetero-dimers, supporting that Ca²⁺ is supplied from in the ER and its binding is required for higher-order assembly in the cells (Fig. 6a,b). Notably, such effect was not observed for LAT1-CD98hc, which does not have calcium binding sites and does not form a super-dimer (Fig. S12f).

Second, we re-acquired all fluorescence images for the Ca²⁺-binding site mutants under a new experimental setup (Figs. 7, 8c and S14). In the original manuscript, we took the images using a regular fluorescent microscope. In this revised version, we renewed all images by using a confocal microscope with a 100x lens and included the PDI and RCAS1 staining as the ER and Golgi apparatus markers, respectively, to observe the subcellular localization at better resolution. The results indicated that the majority of Ca²⁺-site mutants remained at the ER, whereas wild-type and E321K reached the plasma membrane (Figs. 7, S14).

Together, our new results show that $b^{0,+}AT$ -rBAT_c hetero-dimerization is followed by super-dimerization, and the latter step is required for protein trafficking from the ER to Golgi apparatus and the plasma membrane. As already shown in the original manuscript, mutations in the Ca²⁺-binding site (N214A, D284A, and E321A) and the super-dimer interface (D349R/D359R (RR) and R362D/R326D (DD)) dramatically reduce super-dimers (Fig. 5g-h, S9), thus making most proteins retain at the ER (Fig. 7a, S14b). As a result, the mutants showed impaired N-glycan maturation (Fig. 5d-e) and transport function (Fig. 5j). Taken together, we propose the model of rBAT- $b^{0,+}AT$ biogenesis, as shown in Fig. 10.

We changed the following parts:

- Lines 326–409, pages 14–17:

We added a new section “ER Ca²⁺ mediates super-dimerization for system $b^{0,+}$ biogenesis”

- Fig. 6; Fig. S12

- Fig. 5d-j (newly rearranged from the original manuscript)

I would have liked to see localisation data for the different mutations reported in Figure 5d and 5h. Essentially the main novelty in this report are the insights gained into the role of rBAT in trafficking the dimer of heterodimers ([rBAT/b^{0,+}AT]₂) in the cell.

Answer:

Following the reviewer’s comment, we have performed immunofluorescence imaging of several rBAT mutants, including the ones that are involved in Ca²⁺ binding as shown in Fig. 7 (N214A, D284A, E321A) and Fig. S14 (corresponding mutants on a V355C background) as well as cystinuria mutants (Fig. 8c). To further clarify organellar localization, we have included PDI and RCAS1 staining as the ER and Golgi-apparatus markers, respectively. The results clearly showed that all rBAT mutants involved in calcium binding fail to traffic to the plasma membrane and most of the mutant populations are retained in the ER.

Please see detail in the revised manuscript:

- Result lines 393–408, page 17.

- Figs. 7, 8c and S14.

Although I applaud the authors in their structural work, I didn't see anything novel in this study from a structural perspective beyond the previous Sci Adv paper by Yan et al (DOI:

10.1126/sciadv.aay6379), except the identification for a role of calcium in dimer formation, which pertains to the trafficking.

Answer:

We are in the same opinion that our structural description has many overlaps with those presented by Yan et al., Sci Adv, 2019 and Wu et al., PNAS, 2020. However, we also think that some of the key structural details are not fully described in these works, especially those related to Ca²⁺ and type I cystinuria mutation. For example, the Ca²⁺-binding site is in the proximity of Thr216, which causes a common cystinuria mutant T216M. In addition, R365W and M467T are located at the subdomain interfaces, suggesting how these mutations might destabilize the rBAT ectodomain. This structural information is used as a basis for downstream biochemical analyses in our paper, while it was overlooked in both previous studies. Therefore, we think that the structural work is still an important part of our current study.

The EndoH assay is rather low resolution from a mechanistic perspective - essentially is distinguishes those proteins that have trafficked to the PM from those that haven't. It doesn't tell us anything about what is happening in the cell. For example, do some of these mutants get trapped in different parts of the ER or Golgi? Are they degraded at different rates? I don't think this is an unfair comment, as the authors refer to degradation in Figure 7. But this rather a catch all term - what route of degradation, on what time scale. Answers to these questions would address a mechanistic view of the phenotypes seen in patients heterozygous for trafficking mutations.

Answer:

To address the reviewer's main concern on the molecular mechanism of the wild-type protein and pathological mechanisms of the mutants, we have performed series of biochemical experiments to clarify mechanisms of protein biogenesis, trafficking and degradation. The details are described below:

Regarding the first question "do some of these mutants get trapped in different parts of the ER or Golgi?", we have now performed confocal fluorescence imaging by adding ER marker (PDI) and Golgi-apparatus marker (RCAS1) and analyzed protein localization at higher resolution (Figs. 7 and S14). The results answer that all rBAT calcium-binding mutants (N214A, D284A, and E321A) mainly retain at the ER but not at the Golgi apparatus, and the cation-compensating E321K recovers the plasma membrane localization.

Regarding the question on protein degradation, we have performed cycloheximide-chase assays to monitor the stability of different assembly or maturation intermediates. The results showed that rBAT monomer and core-glycosylated $b^{0,+}AT$ -rBAT_c heterodimer are degraded faster (continuously degraded since 1 h and ~70% degradation during 6 h) than the super-dimer (stable during 1-6 h) (Fig. 6f-h). Together with the fluorescence imaging results, these results show that the rBAT monomer and core-glycosylated rBAT- $b^{0,+}AT$ are trapped in the ER and are subject to the ER-associated degradation processes, as described by Bartoccioni et al., Hum Mol Genet 2008 (Ref. 12). Meanwhile, those super-dimers that reached the plasma membrane are relatively stable (Fig. 6f-h).

We also deleted the word “degradation” from the legends for Fig. 10 (originally Fig. 7), as we did not perform any particular experiment to determine which degradation pathway the cystinuria mutants are subject to. In the new Fig. 10, we rather emphasize those steps that the known mutants would affect. For example, we found that T216M behaves similarly to the Ca^{2+} -binding site mutants which disrupt the super-dimer formation. Thus, T216M impairs the system $b^{0,+}$ biogenesis at the ER. R365W and M467T mutations partially disrupt super-dimerization and protein maturation, granting our suggestion that both mutations defect protein biogenesis and/or protein stability at both ER and Golgi apparatus.

Please see detail in the revised manuscript:

- Lines 326–409, pages 14–17
- Figs. 6, 7, S14

We agree that providing “a mechanistic view of the phenotypes seen in patients heterozygous for trafficking mutations” would be very interesting. To study this, one would have to generate the hetero-complex formed by wild-type and mutants and observe cellular behavior at a single molecule level. However, in our current setup, it is technically challenging to do so, and it goes out of the scope of the current manuscript. We would like to leave this very interesting topic for future.

I felt the trafficking analysis needed more analysis and work. The current results definitely prove that rBAT dimerisation is required for the efficient transport of ([rBAT/b0,+AT]2) to the PM, however, that was already known (see Fernandez et al. JBC 281, p36 (2006). Unless the authors can argue that the role of calcium here is regulatory?

Answer:

It is true that the role of rBAT dimerization in protein trafficking has been elegantly shown by Fernandez et al., JBC, 2006 and subsequent works, and our current study is clearly building upon these findings. The key contribution from our study is the finding that Ca^{2+} binds early in biogenesis and it strengthens the rBAT homo-dimerization interface, which was not known before. Together with our original data in Fig. 5d-j and the additional data from the pulse-chase experiment (Fig. 6) and thapsigargin treatment (Fig. S12), current evidences support the role of the ER calcium on super-dimerization ([rBAT/b0,+AT]2), which is indispensably required for the protein trafficking from the ER to the Golgi apparatus and plasma membrane (Fig. 7). We do not think Ca^{2+} here is regulatory, unlike calmodulin and other fast Ca^{2+} responsive elements, since Ca^{2+} binding appears to be very strong and irreversible (Fig. S13).

Please see detail in the revised manuscript:

- Lines 348 – 363, page 15
- Figs. 6, 7, 8, S12 and S13

Bartoccioni et al Human Mol Genet 2008 proposed a biogenesis model in which the initial step is $\text{b}^{0,+}\text{AT}$ -rBAT dimerization, followed by the rBAT ectodomain folding. Our pulse-chase experiment (Fig. 6a,b) and biochemical analyses of $\text{b}^{0,+}\text{AT}$ -L89P and rBAT-G105R (Figs. 8a-c, S9-10) agree to Bartoccioni et al and Rius et al. JBC 2012 that the dimerization of rBAT and $\text{b}^{0,+}\text{AT}$ is initially required and could be counted as an earlier step in the biogenesis. By using ER calcium inhibition for pulse chase assay (Fig. 6a), we show that the ER calcium is required for super-dimer formation, supported by the ER localization of Ca^{2+} site mutants (Fig. 7). This Ca^{2+} -dependent super-dimerization is indispensable for the protein complex to leave the ER and could be counted as the later step in the biogenesis. Incorporating these findings, we have modified Fig. 10 (formerly Fig. 5).

At the very least I would also like the authors to run FSEC analyses on the mutants used to demonstrate the proteins are still stable and that the negative glycosylation regult is not due to protein misfolding/aggregation as opposed to abherrent trafficking.

Answer:

We performed FSEC analyses on those mutants that were found to disrupt super-dimeric interactions. The result showed that most mutants were unstable when solubilized in detergent solution (Fig. S13), suggesting that super-dimerization is also important for keeping protein stability. It should be noted that detergent extraction is generally very harsh to proteins, and proteins being able to fold does not necessarily mean they show a monodisperse peak in FSEC.

Supporting this, when we subject cell lysates directly to SDS-PAGE, those mutant bands are detected, which would not be the case if the mutants had completely failed to express (except for b^{0,+}AT G105R). Therefore, these results suggest that mutants do decrease protein stability, in addition to causing the aberrant trafficking.

Please see detail in the revised manuscript:

- Lines 358 – 363, page 15.

- Fig. S13

The insights gained from the transport assays is also novel and well executed. The authors main point here is that Asp233 and Asn236 are important for the selectivity of amino acids in this system. This observation builds on previous work in the literature. In addition. the authors suggest that system b^{0,+} is more similar to y⁺L than system L or the CATs, which is functionally interesting. The assay data is good, however, I would like to see SDS-PAGE gels for each of the mutants to show that reconstitution was the same for each sample analysed in Figure 6g.

Answer:

We thank the reviewer for positive comments regarding the insights into substrate selectivity. In the original manuscript, we performed uptake assays for cationic (Orn), neutral (Tyr) and small (Ala) amino acid substrates. To complete the story of substrate specificity, we have additionally performed an assay for cystine, which is one of the main physiological substrates of b^{0,+}AT and is accounted for cystinuria in the impaired proteins (Fig. 9j). Our results indicated that cystine transport in D233 and N236 mutants adopts the recognition of both cationic (Orn) and neutral (Tyr) amino acids. Cystine transport and amino acid alignment support our speculation in the original manuscript that b^{0,+}AT utilizes multiple residues for the substrate recognition and these residues are interchangeable within the substrate binding pocket.

Please see detail in the revised manuscript:

- Lines 499 – 506, page 21

- Fig. 9j

As for SDS-PAGE of mutant proteins, we deeply apologize that we made a serious mistake in the figure legends for Figure 5, which incorrectly read:

“Uptake of L-[14C]-Orn by proteoliposomes reconstituted with b^{0,+}AT-rBAT or its mutants.”

All uptake experiments except for Fig. 1 were done in cell-based systems, which would show combined effects of protein mis-localization and low catalytic activity. We regret that this typo has caused misinterpretation in the previous manuscript. It was probably introduced when we copied the legends from Fig. 1 to Fig. 5, which both concerns transport activity, and it escaped our notice while reviewing the manuscript. In the revised manuscript, we have corrected all figure legends and explained more clearly that we used HEK293 cell transfection. Nevertheless, our conclusion does not change, as these mutations were hypothesized to affect the localization of b⁰⁺AT-rBAT rather than the catalytic activity.

- In Fig. 9 legends:

“Uptake of L-[¹⁴C]-ornithine by proteoliposomes reconstituted with in HEK293 cells transfected with b⁰⁺AT-rBAT or its mutants”

Reviewer #2 (Remarks to the Author):

The manuscript submitted by Lee and colleagues uncovers one of the molecular mechanisms leading to Cystinuria as a consequence of congenital mutations localized in rBAT, the beta subunit of the amino acid transporting system b⁰,⁺, composed by the heterodimerization of rBAT and the amino acid transporter b⁰,⁺ AT. As in the previous structures of rBAT-b⁰,⁺ AT, here the authors identified a Ca²⁺ ion coordinated by residues located at the interface that bridges two r-BAT/b⁰,⁺ AT heterodimers, thus stabilizing a supercomplex formed by the two heterodimers. Previous studies (including this one) demonstrated that the formation of this supercomplex is pivotal for protein maturation and trafficking. In this work, the authors demonstrate using functional and biochemical experiments that Cystinuria mutations that perturb this Ca²⁺ binding site at the dimer-dimer interface also impair both, the formation of this supercomplex and protein maturation resulting in a defect of trafficking of the complex to the final membrane destination.

This work provides a step ahead on our understanding of the molecular mechanisms leading to Cystinuria. Combining structural and biochemical data, the authors establish solid links between rBAT-b⁰,⁺ AT maturation and r-BAT/b⁰,⁺ AT supercomplex formation where the Ca²⁺-mediated r-BAT-rBAT dimerization plays a leading role, thus uncovering the molecular mechanism leading to Cystinuria of rBAT variants found in Cystinuria patients. This information is also relevant for future development of therapeutic strategies to cure Cystinuria. Moreover, the conclusions expressed by the authors are well supported by the experimental data. Therefore, in my opinion, this manuscript reaches the standards for publication in Nature Communication and therefore I recommend it for publication. Nevertheless, some questions and remarks need to be addressed by the authors before the final acceptance of this manuscript.

Answer:

We thank the reviewer for recognizing the findings in our paper and recommending it for publication, while also providing critical comments and questions. Below we answer all comments one by one. Please also see our answers to another reviewer, in which we detail additional experiments we have performed, such as pulse-chase assay, cycloheximide-chase assay, and fluorescence imaging to study subcellular localization.

1. It is important that the authors discuss further about the overall conformation of the ovine r-BAT/b⁰,⁺ AT structure obtained in their work and compare it with the previous two structures of human r-BAT/b⁰,⁺ AT. For instance, is b⁰,⁺ AT in the same conformation as the previous structures ?

Answer:

We have now added a new panel in Fig. S6c comparing our ovine b⁰⁺AT structure with two recently published structures of human b⁰⁺AT. The superposition of the three structures shows that they are essentially in the same conformation. We also added texts discussing this point:

- Lines 133 – 134, page 6:

“Superimposition of ovine b⁰⁺AT with two structures of human b⁰⁺AT recently reported (14, 15) shows that our structures are in the same conformation as the previous ones.”

- Line 445, page 19:

“Our b⁰⁺AT structure adopts the inward-facing conformation without substrate-bound.”

2. To better understand the objective of the EndoH experiments, the authors need to explain better the assay. For instance, fully mature glycosylated protein is resistant to EndoH treatment.

Answer:

Following the reviewer’s suggestion, we have included the explanation for the Endo H assay:

- Lines 266 – 269, page 12:

“In this assay, fully mature glycosylated proteins that reached the Golgi apparatus are resistant to Endo H treatment, whereas core glycosylated ones that are trapped in the ER remain Endo H sensitive, thereby allowing us to investigate protein maturation as a proxy for protein trafficking.”

3. Is the formation of the supercomplex affected at some extent by the presence of the fused fluorescent proteins ?

Answer:

To test possible effects of fluorescence tags, we have performed non-reducing SDS-PAGE and functional analyses of the cells transfected with and without fluorescent fusion proteins (Fig. S11). The results indicate that both EGFP and mCherry tags do not interfere with super-dimerization and do not alter the transport function of the proteins. We also evaluated the expression patterns in HEK293 and HeLa cells since we used both cell types in our experiments, and the results show that cell types also do not alter the outcome of the experiments.

Please see detail in the revised manuscript:

- lines 306 – 307, page 13:

“Prior to the analysis, we confirmed that fluorescence tags or host cell types did not significantly alter protein function or complex formation.”

- Fig. S11

4. In line 269 the authors state that a decrease in maturation due to mutations at positions N214, D284 or E321 leads to a decrease on Ca²⁺ binding. While theoretically these mutations could disturb the Ca²⁺ binding site, the authors do not provide experimental evidences of this (e.g. a Ca²⁺ binding assay ?). For instance, have the authors tested the effect of Ca²⁺ chelating agents on the supercomplex formation after protein purification? . Please, can you comment ?

Answer:

Following the reviewer’s suggestion, we have now tested the effect of Ca²⁺ chelating agent EGTA on the stability of rBAT/b^{0,+}AT in detergent solution. The result showed that 10 mM EGTA did not affect the size or stability of the complex (Fig. S13a). It suggests that Ca²⁺ is strongly bound to rBAT and is not removable by external chelators like EGTA once the superdimer is formed. This is in line with our model that Ca²⁺ is acquired early during biogenesis and is important for cellular functions, rather than being a regulatory factor.

Given the above negative results in stripping off Ca²⁺ by EGTA, we analyzed the effects of a calcium pump inhibitor Thapsigargin (Tg), which limits the ER Ca²⁺ influx, in pulse-chase assay (Fig. 6a,b) and at saturated expression level (Fig. S12). We monitored the behaviors of both super-dimers and other assembly intermediates. The results showed that Tg strongly inhibits super-dimerization of b^{0,+}AT–rBAT (Fig. 6b), while such effect was not observed in LAT1–CD98hc (Fig. S12f), which does not have a Ca²⁺ binding site. Although it cannot be excluded that Ca²⁺ inhibition may also affect other quality control proteins residing in the ER (such as glycan processing enzymes and disulfide isomerase), this result is supportive of our model that Ca²⁺ is incorporated during early biogenesis and is required for protein higher-order assembly.

These results are consistent with our cross-linking assays and new high-resolution fluorescence images, which showed that rBAT Ca²⁺-binding site mutants (N214A, D284A and E321A) fail to form super-dimers (Figs. 5g, S12-13b) and retain in the ER (Fig. 7a). Accordingly, the mutants are decreased in N-glycan maturation which is the Golgi-apparatus-resided stage (Fig. 5d) and result in no transport function (Fig. 5j).

Please see detail in the revised manuscript:

- Lines 326–409, pages 14–17:

We added a new section “ER Ca²⁺ mediates super-dimerization for system b^{0,+} biogenesis”

- Figs. 5d, g, j (newly rearranged from the original manuscript)

- Fig. 6

- Fig. S14

5. Line 289. What are the experimental evidences supporting that G105R mutation in b^{0,+} AT leads to improper protein folding ?.

Answer:

We apologize about the unclear explanation on G105R. The reason for this statement is that we had observed in SDS-PAGE that G105R shows extremely low expression (Fig. S9b: lane 6 rBAT-WT + b^{0,+}AT-G105R). Since rBAT requires b^{0,+}AT for oxidative folding, which has been shown previously (Ref. 22: Rius et al. JBC 2012), this in turn resulted in immature rBAT. For clarity, we have changed the word “folding” to “expression” and added figure reference to the concerned text.

To further address the effect of G105R, we have additionally performed fluorescent imaging to observe its localization, as shown in Fig. 8c. The result showed that b^{0,+}AT-G105R does not translocate to the plasma membrane and most of the proteins are retained in the ER. In fact, the expression level of mCherry-b^{0,+}AT-G105R in the fluorescent imaging is very low, which is consistent with the expression level on the SDS-PAGE in Fig. S9b. Taken together, we suggest that the G105R affects the b^{0,+}AT folding, resulting in b^{0,+}AT degradation, no rBAT maturation and its trapping in the ER.

Please see detail in the revised manuscript:

- Lines 422 – 424, page 18:

“b^{0,+}AT G105R showed almost no expression (Fig. S9b), causing to the loss of rBAT maturation (Fig. 8b: lanes 2-3), consistent with previous observation that b^{0,+}AT is required for oxidative folding and maturation of the rBAT ectodomain.”

- Lines 534-536, page 22:

“Interestingly, b^{0,+}AT-G105R showed little expression and resulted in no rBAT maturation (Fig. 10a), suggesting that some of non-type I mutations are, at the molecular level, protein-folding defects.”

- Fig. 8b,c, S9b

6. Lines 310-312 and Figure 5h. In lanes 3-5 in Figure 5h, the intensities of the bands of these mutants are much weaker than WT, even at the gel bands assigned to each monomeric protein (also see in Supplemental Figure S10). Therefore, other mechanisms other than the disruption of the superdimer could be envisaged to explain the impairment of protein expression caused by these mutations ?. If this is the case, the authors should comment this in the main text.

Answer:

This is an important point. In a simplified mechanism, we put super-dimerization as the key difference between the behaviors of wild-type and mutant proteins. However, the band intensities in SDS-PAGE do suggest that these mutations may affect protein expression to some extent and thereby change the effective protein amount. The reviewer's comment have encouraged us to perform further analysis of b^{0,+}AT-rBAT biogenesis, as shown below.

First, we performed pulse-chase experiment to observe protein biogenesis. The result showed that the Ca²⁺-binding mutants (D284A and E321A) do not significantly alter the amount of core-glycosylated rBAT monomers (rBAT_c) or b^{0,+}AT-rBAT_c heterodimers, but markedly disrupt super-dimer formation (Fig. 6b: lanes 1-6). Inhibition of the ER Ca²⁺ flux by thapsigargin (ER calcium pump (SERCA) inhibitor) abolished the super-dimer formation in both rBAT-WT and all mutants, and instead generated an aggregation at the top of the gel (Figs. 6b-e). Similar phenomena were found at prolonged expression (2-day). The Ca²⁺-binding site mutants (N214A, D284A and E321A) hardly produced super-dimers and thapsigargin treatment caused aggregation (Fig. S12).

In the new cycloheximide-chase experiment, we transfected equal amounts of rBAT Ca²⁺-binding site mutation plasmids and could observe the protein assembly intermediates (Fig. 6b-d). In the case of D284A and E321A, we observed similar amounts of rBAT monomers but different amounts of super-dimers, indicating that the Ca²⁺-binding mutants do not affect protein expression (Fig. 6b: chase 0 h). However, over time, effective amounts of total rBAT would gradually change as the incomplete assembly intermediates would degrade faster, which resulted in the different intensity of the bands in SDS-PAGE.

In conclusion, we propose that impairment of Ca²⁺ binding results in the failure of super-dimerization and this in turn generates protein aggregation (and probably further go to the degradation process) but does not significantly affect the rBAT expression level itself.

Please see detail in the revised manuscript:

- Lines 326–409, pages 14–17:

A new section “ER Ca²⁺ mediates super-dimerization for system b_{0,+} biogenesis”

- Figs 6b-e, S12

7. Line 336 and 354. These sentences should be reconsidered as this does not seem true for the L89P mutant.

Answer:

Following the reviewer’s suggestion, we have rewritten the paragraph in the Results. We have included more experiments providing mechanistic insights into b^{0,+}AT-rBAT biogenesis.

In brief, we demonstrated two-step biogenesis: the initial step including the heteromeric assembly of b^{0,+}AT-rBAT_c and the following step including the Ca²⁺-mediated super-dimerization. Both steps are required for the ER quality control. The L89P impaired the initial assembly of b^{0,+}AT-rBAT in the biogenesis, since the L89P mutant appears to be inefficient in forming the heterodimer (Figs. 8b, S9). As a result, the mutant retained in ER (Fig. 8c) and displayed no maturation (Fig. 8a) and no transport function (Fig. 8d).

Please see detail in the revised manuscript:

- Lines 417 – 421, page 18:

“Consistent with the previous studies (11, 39), rBAT L89P diminished the amount of b_{0,+}AT-rBAT heterodimer and thus also the super-dimer (Fig. 8a: lane 3), accompanied by the low level of N-glycan maturation and the ER localization, supporting that L89P mutant impairs the heteromeric interaction between b_{0,+}AT and rBAT and thereby decrease the functional complexes.”

Minor points

Line 64 . Please, give more details concerning the sentence “...the insertion of 110 residues...”. In particular, in what part of the protein is found this extra region ?.

Answer:

Most of the additional residues found in rBAT are located in domain B loops and the C-terminal region. We have included this information in the revised manuscript:

- Lines 62 – 63, page 3:

“..... the insertion of a total of ~100 residues in domain B loops and the C-terminal region.”

Line 86 . The term “trafficking disease” could be misleading. Can the authors confirm that this term is commonly used in the field ? .

Answer:

Thank you for pointing this out. Although we do find the word “trafficking disease” in some literature (mostly as “protein-trafficking disease” or “membrane-trafficking disease”), following the reviewer’s suggestion, we have corrected the words in the revised manuscript.

- Lines 84-86:

“Furthermore, we found that the loss of higher-order assembly is correlated with some type I mutations, revealing a previously unknown link of protein oligomerization and the disease.”

Line 92. Please change “ good solution behavior in detergent” by “good behavior in detergent solution”

Answer:

We appreciate the reviewer for careful reading. We have changed the sentence following the reviewer pointed as shown in lines 91 – 93, page 5:

“...and showed good behavior in detergent solution...”

Line 374. In the absence of an experimental evidence I suggest to change “...indicating that...” by “suggesting that..”.

Answer:

Following the reviewer's point, we have changed the sentence as shown in lines 458–459, page 19:

“..., indicating that one acidic residue is sufficient for recognizing cationic substrates in system b^{0+} .”

Fig 7a. There is a missing reference.

Answer:

Thank you for pointing this out. We corrected the reference in legends for Fig. 10 (former Fig. 7). In the original manuscript, we did not have strong biochemical evidences to clarify the biogenesis mechanism of $b^{0,+}$ AT-rBAT in ER. Encouraged by the valuable comments from the reviewers, we have now included pulse-chase assay, cycloheximide-chase assay, thapsigargin (ER Calcium pump inhibitor) treatment experiment and fluorescence imaging to explain the biogenesis of $b^{0,+}$ AT-rBAT. These results support the model in Fig. 10a (former Fig. 7a as the reviewer commented). In addition, we also include the Reference to Bartoccioni et al. who studied the initial assembly of $b^{0,+}$ AT-rBAT and discovered some of the critical steps, including heteromeric assembly and oxidative folding of rBAT.

Line 459. I suggest to describe AdiC and add a reference.

Answer:

We added brief descriptions for functional and structural aspects of AdiC and BasC.

- Lines 548–552, page 23:

“The comparison of outward- and inward-facing structures of bacterial homologs of the SLC7 family, known as AdiC and BasC (49, 50) shows that, upon the inward-to-outward structural transition, EL4b dissociates from TM3, and EL4a undergoes upward movement to widen the transport pathway (Fig. S16c,d).”

REVIEWERS' COMMENTS

Reviewer #1 (Remarks to the Author):

The authors have addressed all my concerns with new experimental evidence and improved data acquisition. The conclusions surrounding the role of calcium in 'super-dimer' formation and the implications for rBAT biogenesis are now substantially more robust in my opinion. The current version of the manuscript now adds considerable advances to the field of SLC7 biology and membrane protein biogenesis. I applaud the authors for a very nice study.

Reviewer #2 (Remarks to the Author):

In this new and revised version, Lee and colleagues have addressed satisfactorily the different questions I raised in my previous report. The new experimental data added in the revised version, besides addressing the reviewers' concerns, gives a better support of the author's statements and working models. Therefore, I recommend to accept this new version for publication.

Response to referees

Reviewer #1 (Remarks to the Author):

The authors have addressed all my concerns with new experimental evidence and improved data acquisition. The conclusions surrounding the role of calcium in 'super-dimer' formation and the implications for rBAT biogenesis are now substantially more robust in my opinion. The current version of the manuscript now adds considerable advances to the field of SLC7 biology and membrane protein biogenesis. I applaud the authors for a very nice study.

Answer:

Thank you very much for positive comments on the revised manuscript and commenting on implications of this manuscript to the field of SLC7 biology. We appreciate that the constructive suggestions from the reviewer have improved our manuscript significantly.

Reviewer #2 (Remarks to the Author):

In this new and revised version, Lee and colleagues have addressed satisfactorily the different questions I raised in my previous report. The new experimental data added in the revised version, besides addressing the reviewers' concerns, gives a better support of the author's statements and working models. Therefore, I recommend to accept this new version for publication.

Answer:

Thank you very much for positive comments and recommendation to accept the new version for publication. The questions raised in the previous report inspired us to perform additional experiments, which now support working models for system b^{0,+} biogenesis. We appreciate again for your effort and time in reviewing our manuscript.